# Learning Anisotropic Value Geometry with Finsler Reinforcement Learning

**Jumman Hossain** [1]   **Nirmalya Roy** [1]

## Abstract

We introduce **Finslerian Reinforcement Learning (FiRL)**, an RL framework that makes directional costs explicit and improves robustness to tail risk. FiRL incorporates a *Finsler metric* into the locomotion cost, expressing effort as $F(x, v)$ that depends on the state $x$ and motion $v$, so it can capture uphill versus downhill asymmetry, lateral slip, and other direction-dependent effects. To handle rare but catastrophic outcomes, FiRL optimizes a Conditional Value-at-Risk ($\text{CVaR}_\alpha$) objective. We derive the corresponding risk-sensitive Bellman equation and show that the resulting CVaR–Finsler Bellman operator is a $\gamma$-contraction. This guarantees a unique fixed-point value function, while the underlying Finsler cost induces an asymmetric path cost $d_F$ that satisfies a triangle inequality despite directional asymmetry. We then develop a FiRL actor–critic algorithm to learn policies under this anisotropic, risk-averse objective. Across simulation benchmarks and real-world robot trials, FiRL demonstrates safer and more energy-efficient locomotion behavior than strong baselines such as risk-neutral PPO. For instance, on a $12°$ sloped Hopper task, FiRL reduces worst-case ($\text{CVaR}_{0.1}$) impact forces by over 35% and total energy cost by 15%, while also improving success rate.

**Project Page:** https://pralgomathic.github.io/FiRL/

## 1. Introduction

Legged robots offer unique advantages in navigating unstructured terrains, but they also face significant challenges in energy management and safety (Miki et al., 2022; Fu et al., 2021; Kumar et al., 2021; Fu et al., 2022a; Mahankali et al., 2024). Recent advances in reinforcement learning (RL)

have produced agile locomotion controllers for complex robots (Hwangbo et al., 2019; Rudin et al., 2022; Zhuang et al., 2023). However, standard RL formulations often assume *isotropic* (direction-agnostic) cost or reward functions and optimize solely for expected return (Rudin et al., 2022; Hwangbo et al., 2019; Lyu et al., 2024; Xu et al., 2024). This overlooks two crucial aspects of real-world locomotion: (1) **Directional anisotropy in cost** – moving in different directions can incur vastly different energy expenditures and wear (Fu et al., 2022b)(e.g., climbing uphill requires far more work than going downhill, sharp turns can waste momentum, descending steps quickly can be dangerous); and (2) **Tail-risk sensitivity** – rare but high-cost events such as slipping, falling, or hardware strain must be minimized for safe deployment. By ignoring anisotropic effort and focusing only on average performance, conventional RL policies may appear efficient on flat terrain but perform poorly on slopes or under disturbances, suffering catastrophic failures in the worst cases (Shi et al., 2024; Du et al., 2023).

Recent work on geometry-aware RL has explored embedding inductive biases about distance and dynamics into value functions. For example, Wang et al. (2023) enforce a *quasi-metric* structure on value estimates to satisfy triangle inequality and asymmetry, improving generalization in goal-reaching tasks. Others have applied *Riemannian* geometry to RL, e.g. shaping policy updates or value functions using symmetric metrics (Wang et al., 2020; Kakade, 2001; Vu & Slavakis, 2024). These approaches demonstrate the benefit of geometric priors, but they either assume simplified settings (constant speeds, discrete goals in quasi-metric RL) or impose symmetric structures that cannot capture direction-specific costs. Meanwhile, in risk-sensitive and safe RL, methods like CVaR optimization (Chow et al., 2015; Tamar et al., 2015) and distributional RL (Lim & Malik, 2022) bias the training towards safer outcomes by focusing on the worst-case returns. Notably, Schneider et al. (2024) apply distributional RL to quadrupedal locomotion, showing that risk-aware training can yield cautious gait policies on a real robot. While earlier methods have explored reward shaping for anisotropy and CVaR-based risk objectives (Ying et al., 2022) separately, the integration of direction-dependent costs with explicit tail-risk optimization has not been addressed.

In this paper, we propose **Finslerian Reinforcement Learn-**

---

[1]Department of Information Systems, University of Maryland, Baltimore County, USA. Correspondence to: Jumman Hossain <jumman.hossain@umbc.edu>.

*Proceedings of the 43$^{rd}$ International Conference on Machine Learning*, Seoul, South Korea. PMLR 306, 2026. Copyright 2026 by the author(s).

**ing (FiRL)**, a novel framework that integrates differential geometry with risk-sensitive RL for legged locomotion. FiRL introduces a *Finsler metric* into the cost function, allowing the agent to account for direction-dependent effort: for instance, uphill moves incur higher instantaneous cost than downhill moves, and lateral motions incur frictional penalties (Fig. 1). Simultaneously, FiRL optimizes a *Conditional Value-at-Risk (CVaR)* objective, which biases learning towards minimizing the worst-case outcomes (the tail of the cost distribution) rather than just the mean. By integrating these components, FiRL produces policies that proactively avoid energetically costly maneuvers *and* reduce the probability of catastrophic failures.

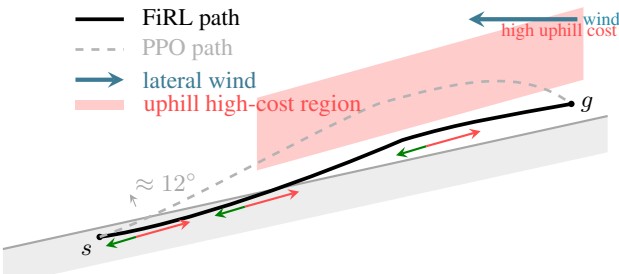

*Figure 1.* **FiRL vs. PPO on a slope with lateral wind.** The Finsler cost assigns higher cost to uphill and lateral motion. FiRL chooses a safer, lower-cost route around the high-cost uphill region, while PPO follows a more direct but riskier trajectory.

**Our main contributions**:

- **Finsler Cost with Tail-Risk Objective**: We propose a new per-step cost formulation using a Finsler metric $F(x, v)$ that depends on the agent's state $x$ and motion direction $v$. This formulation explicitly embeds anisotropic energetics into the cost (rather than treating it as a heuristic), and we optimize a CVaR (Conditional Value-at-Risk) criterion of the cumulative cost. By integrating anisotropic geometry directly into the cost model along with a tail-risk objective, our approach can capture direction-dependent energy penalties in a principled way.

- **Risk-Sensitive Bellman Operator & Induced Geometry**: We derive a CVaR–Finsler Bellman operator for our risk-sensitive setting and prove that it is a $\gamma$-contraction under mild boundedness conditions. This theoretical result ensures the existence of a unique fixed point (optimal value function) and standard convergence guarantees for dynamic programming. Moreover, we show that the local Finsler cost induces an asymmetric path cost $d_F$ on the state space that satisfies a triangle inequality while allowing directional asymmetry. This provides a geometric interpretation of the learned risk-sensitive behavior without requiring

the discounted value function itself to satisfy a triangle inequality.

- **FiRL–AC Algorithm (Actor–Critic)**: We develop FiRL–AC, a new actor–critic learning algorithm that incorporates the anisotropic Finsler cost and CVaR objective into practical reinforcement learning. In FiRL–AC, the per-step cost uses $F(x, v)$ to penalize movements directionally, and the critic is trained with CVaR-based targets to focus on tail outcomes. To rigorously evaluate the impact of anisotropic cost vs. standard cost and CVaR vs. expected cost, we perform a controlled 2×2 ablation study with matched hyperparameters.

## 2. Related Work

**Geometry and manifold methods in learning and control.** Manifold structure has been widely used to model data and improve representation learning (Lin & Zha, 2008), and recent work studies how to learn optimizers that operate directly on Riemannian manifolds (Gao et al., 2022; Lin & Zha, 2008). In RL, geometric structure appears in multiple ways. Riemannian metrics have been used to shape policy learning and provide geometry-aware updates (Sun et al., 2024; Wang et al., 2020). A complementary viewpoint places policy optimization in the space of probability measures and interprets updates through Wasserstein gradient flows (Zhang et al., 2018; Ziesche & Rozo, 2023). In robotics, Riemannian motion policies provide a modular geometric language for motion generation and policy composition (Ratliff et al., 2018). These methods provide strong inductive bias, but they typically assume symmetric geometry. Recent quasi-potential reinforcement learning studies asymmetric traversal costs through structured value representations (Hossain & Roy, 2025). However, FiRL works at the level of local locomotion mechanics, where asymmetry arises from physical effects such as uphill effort, downhill motion, lateral slip, and contact risk. It captures these effects with a Finsler cost and optimizes the resulting anisotropic objective using dynamic CVaR for tail-risk-sensitive control (Bao et al., 2000; Ratliff et al., 2021).

**Risk and anisotropy in legged locomotion.** Modern locomotion RL systems often rely on reward designs that combine forward progress with control penalties and contact terms (Tassa et al., 2018), which do not explicitly encode direction-dependent effort. Energy-focused shaping can encourage efficient gaits (Fu et al., 2022b; Liang et al., 2025), but these terms are commonly symmetric with respect to motion direction. In parallel, risk-aware locomotion work shows that tailoring objectives to the tail of the return distribution can improve safety and robustness (Schneider et al., 2024; Shi et al., 2024; Zeng & Dixit, 2025). FiRL targets the intersection: it encodes directional asymmetry through a Finsler metric $F(x, v)$ and directly optimizes a tail-risk

objective, aiming for policies that are both energy-aware and conservative about rare, damaging outcomes. We discuss additional related work in **Appendix B**.

## 3. Methodology

### 3.1. Problem Formulation and Preliminaries

We consider a standard Markov Decision Process (MDP) formalism for the locomotion task: states $x \in \mathcal{X}$, actions $u \in \mathcal{U}$, transition dynamics $x' = f(x, u)$ (possibly stochastic), and a discount factor $\gamma \in [0, 1)$. However, instead of a conventional scalar reward, we define a *state-action cost* via a Finsler metric $F(x, v)$, where $v$ represents the *motion vector* (e.g., the instantaneous velocity) induced by taking action $u$ in state $x$. Intuitively, $F(x, v)$ measures the **effort or "distance"** incurred by moving with velocity $v$ at state $x$; it generalizes the notion of energy expenditure or risk cost for a small step. We first introduce the components of our Finslerian cost function for locomotion. In our design (Fig. 8), $F(x, v)$ comprises three terms capturing different aspects of motion cost:

**(1) Kinetic energy term $F_{\text{energy}}$:** This term accounts for the basic dynamic effort of movement. We define it as

$$F_{\text{energy}}(x, v) = \sqrt{v^\top M(x)\, v}\,, \qquad (1)$$

where $M(x)$ is a positive-definite weight (or inertia) matrix that can vary with state. In our design, $M(x)$ assigns larger weights to motion along certain axes that are energetically expensive (for example, vertical movements against gravity, or rotations of the body orientation). $F_{\text{energy}}$ is symmetric in $v$ (since it depends on $v$ quadratically) and ensures that faster motions or motions in "heavier" directions yield higher cost.

**(2) Uphill drift term $F_{\text{drift}}$:** To impose an asymmetry between uphill (against gravity) and downhill motion, we add a drift term

$$F_{\text{drift}}(x, v) = \beta(x) \max\!\big(0,\, n(x)^\top v\big)\,, \qquad (2)$$

where $n(x)$ is the unit *uphill* direction at state $x$, defined as the projection of $-g$ onto the local terrain tangent plane (normalized), and $\beta(x) \geq 0$ is a state-dependent weight that increases with the local incline. The $\max(0, \cdot)$ ensures that only motion *against gravity* incurs additional cost: when $n(x)^\top v > 0$ the agent moves uphill and pays a drift penalty, while downhill motion yields $n(x)^\top v < 0$ and contributes zero. In this way, $\beta(x)$ scales the extra effort required to climb—steeper terrain produces larger $\beta(x)$ and thus higher uphill cost, whereas moving downhill or on flat ground incurs no drift penalty. This term makes $F$ *asymmetric*, since in general $F(x, v) \neq F(x, -v)$ (climbing up versus down yields different costs).

**(3) Lateral friction term $F_{\text{friction}}$:** Legged robots also experience higher risks and energy loss during lateral or non-

straight movements (e.g., side-stepping, turning sharply can cause skidding or inefficient gaits). We capture this via:

$$F_{\text{friction}}(x, v) = \lambda \,\|v_\perp\|\,, \qquad (3)$$

where $v_\perp$ is the component of $v$ orthogonal to the robot's primary forward heading direction, and $\lambda > 0$ is a constant coefficient. This term is linear in the lateral speed magnitude, penalizing any sideways motion. $F_{\text{friction}}$ breaks isotropy by distinguishing forward vs. lateral movement directions, although it is symmetric with respect to left vs. right ($v_\perp$ enters through its norm). Essentially, $F_{\text{friction}}$ adds a cost for curving or side-slipping, discouraging high-curvature trajectories that could lead to loss of traction.

**Total Finsler Metric:** We combine the above components into the total cost metric as a weighted sum:

$$F(x, v) = w_e\, F_{\text{energy}}(x, v) + w_d\, F_{\text{drift}}(x, v) + w_f\, F_{\text{friction}}(x, v)\,. \qquad (4)$$

with positive weights $w_e, w_d, w_f > 0$ chosen to balance the contributions of each term.[1]

**Definition 3.1** (Finsler Metric). For each fixed state $x$, the function $v \mapsto F(x, v)$ defined by (4) is a *Finsler metric* on the tangent space $T_x \mathcal{X}$. It satisfies: (i) $F(x, v) \geq 0$ for all $v$, and $F(x, v) = 0$ if and only if $v = \mathbf{0}$; (ii) positive homogeneity: $F(x, \lambda v) = \lambda F(x, v)$ for all $\lambda > 0$; (iii) smoothness in $v$ (except potentially at $v = \mathbf{0}$ or points of nondifferentiability due to the max, which can be handled by subgradients). Unlike Riemannian metrics, a Finsler metric need not be symmetric: in general $F(x, v) \neq F(x, -v)$, which allows encoding one-way directional differences (e.g., uphill vs. downhill).

Our chosen $F(x, v)$ indeed satisfies these properties: $F(x, v) > 0$ for $v \neq 0$, it is positively homogeneous (each component is homogeneous of degree 1 in $v$), and it is smooth except where $v_\| = 0$ (the 'kink' at the switch between uphill and downhill; $\beta(x)$ can be made differentiable and we handle the subgradient at $v_\| = 0$ in practice). Importantly, $F$ is asymmetric due to the drift term, but this asymmetry encodes meaningful physical bias (gravity's effect). By construction, integrating $F$ along a trajectory yields a *path cost*:

$$d_F(\tau) = \int_0^1 F\big(\tau(t), \dot{\tau}(t)\big)\, dt\,,$$

for any path $\tau : [0, 1] \to \mathcal{X}$. In Appendix E, we show that $d_F$ defines a valid *quasi-metric* distance on $\mathcal{X}$: it obeys the triangle inequality (so costs are path-consistent) while allowing $d_F(x, y) \neq d_F(y, x)$. This quasi-metric perspective will be revisited in our theoretical results.

---

[1] We use physically motivated nominal weights and share them between tasks unless otherwise stated; Appendix M reports the chosen values and the sensitivity analysis.

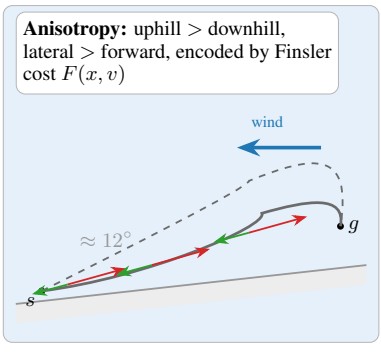 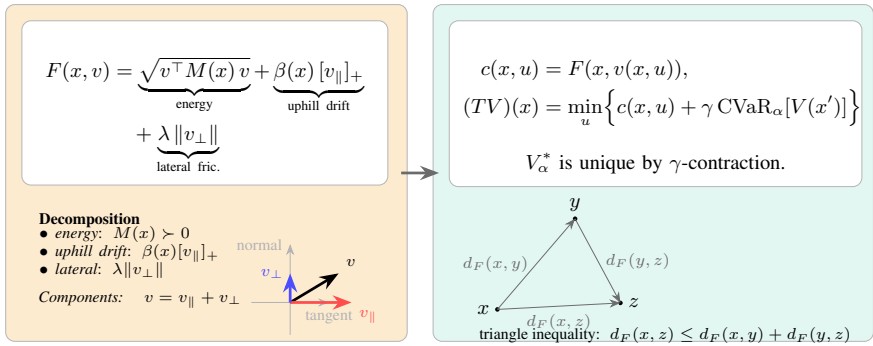

*Figure 2.* **FiRL Overview. Left:** Anisotropic terrain (slope + wind) induces direction-dependent effort: uphill and lateral motions are costly, downhill cheaper. **Middle:** The local Finsler cost $F(x,v)$ decomposes into energy, uphill drift, and lateral friction terms. **Right:** The CVaR Bellman operator combines $F$ with the tail-risk of future value, yielding a unique fixed point $V_\alpha^*$ whose induced path cost $d_F$ satisfies a triangle inequality.

**Dynamic CVaR Risk Objective.** FiRL optimizes a time-consistent, nested CVaR objective rather than a static episodic CVaR of the full trajectory. Let $c_t = F(x_t, v_t)$. The risk-sensitive value is defined recursively through the one-step conditional risk mapping

$$V_\alpha(x) = \min_{u \in \mathcal{U}(x)} \Big\{ c(x,u) + \gamma \,\mathrm{CVaR}_\alpha[V_\alpha(X')] \Big\},$$
$$X' \sim P(\cdot \mid x, u).$$

This dynamic formulation preserves Markovian optimality and matches the Bellman operator used in Section 4.1. We still report empirical $\mathrm{CVaR}_{0.1}$ of rollout costs as an evaluation metric, but the learning objective is the nested Bellman risk objective.

**Cost-only Training Objective.** In the current experiments, we intentionally use a cost-only formulation. The environment defines the task through episode structure, termination conditions, and success metrics, while the optimization objective is the cumulative FiRL cost rather than a mixture of dense task reward and geometric cost. This design isolates the effect of the anisotropic Finsler cost and the CVaR objective. For richer goal-driven tasks, a natural extension is a hierarchical formulation in which a high-level policy optimizes task reward or waypoint selection, while a low-level FiRL controller optimizes direction-dependent cost and tail risk.

## 4. Finslerian CVaR Bellman Equation and FiRL-AC Algorithm

### 4.1. CVaR Bellman Operator and Contraction

We seek an optimal value function under an anisotropic one-step cost and a tail-risk objective. Let $V : \mathcal{X} \to \mathbb{R}$ be any bounded value function. We define the risk-sensitive

Bellman operator $T$ by

$$(TV)(x) = \min_{u \in \mathcal{U}(x)} \Big\{ c(x,u) + \gamma \, \rho_\alpha\big[V(X')\big] \Big\}, \quad (5)$$

where $X' \sim P(\cdot \mid x, u)$ is the next state, $\rho_\alpha[\cdot] = \mathrm{CVaR}_\alpha(\cdot)$, and the one-step cost is

$$c(x,u) = F\big(x, v(x,u)\big).$$

Here $v(x,u)$ denotes the instantaneous base velocity resulting from applying action $u$ at state $x$ (read from the simulator state or approximated by finite differences). The operator in (5) adds the immediate anisotropic cost to the discounted CVaR of the next-state value, and then chooses the action that minimizes this total.

The optimal risk-sensitive value function $V_\alpha^*$ is a fixed point of $T$, i.e.,

$$V_\alpha^*(x) = (TV_\alpha^*)(x) \quad \text{for all } x \in \mathcal{X}.$$

To justify iterative solution methods, we show that $T$ is a contraction under the sup norm. We use two standard properties of CVaR as a coherent risk measure: (i) *monotonicity* (if $Z_1 \le Z_2$ almost surely, then $\rho_\alpha[Z_1] \le \rho_\alpha[Z_2]$), and (ii) *positive homogeneity* ($\rho_\alpha[cZ] = c\,\rho_\alpha[Z]$ for $c \ge 0$). These properties yield the same stability argument as in the risk-neutral case: differences in value functions are discounted by at most $\gamma$ after applying $T$. In Appendix E.1, we formalize this and prove that $T$ is a $\gamma$-contraction in $\| \cdot \|_\infty$, which guarantees a unique fixed point and convergence of value iteration.

**Theorem 4.1** (CVaR–Finsler Bellman Contraction). *Assume the transition dynamics yield bounded costs and that $F(x,v)$ is bounded and Lipschitz in $x$ (ensuring $TV$ maps bounded continuous functions to bounded continuous functions). Then $T$ as defined in (5) is a contraction mapping with factor $\gamma < 1$ in the sup norm. That is, for any two*

value functions $V_1, V_2$,

$$\|TV_1 - TV_2\|_\infty \ \leq \ \gamma \, \|V_1 - V_2\|_\infty \, .$$

*Consequently, $T$ has a unique fixed point $V_\alpha^*$, and iterative application $V^{(k+1)} \leftarrow TV^{(k)}$ converges to $V_\alpha^*$ from any initial $V^{(0)}$. Moreover, $V_\alpha^*$ is the optimal risk-sensitive value function for the MDP (the CVaR-optimal value).*

*Proof.* The proof (detailed in Appendix E) adapts standard contraction arguments to the CVaR case. For any two value functions $V_1, V_2$ and any state $x$, using the definition (5) we get:

$$(TV_1)(x) - (TV_2)(x) = \min_u \Big\{ F(x, f(x,u)) + \gamma \, \rho_\alpha[V_1(x')] \Big\}$$
$$- \min_u \Big\{ F(x, f(x,u)) + \gamma \, \rho_\alpha[V_2(x')] \Big\}$$
$$\leq \gamma \max_u \Big( \rho_\alpha[V_1(x')] - \rho_\alpha[V_2(x')] \Big) \, .$$

 since the immediate cost terms cancel for a given action. By CVaR's monotonicity, $\rho_\alpha[V_1(x')] - \rho_\alpha[V_2(x')] \leq \rho_\alpha[V_1(x') - V_2(x')] \leq \|V_1 - V_2\|_\infty$. Thus $(TV_1)(x) - (TV_2)(x) \leq \gamma \|V_1 - V_2\|_\infty$. Reversing the roles of $V_1, V_2$ yields a symmetric bound in the other direction. Therefore $|(TV_1)(x) - (TV_2)(x)| \leq \gamma \|V_1 - V_2\|_\infty$ for all $x$, proving the contraction. Uniqueness of the fixed point and convergence follow by Banach's fixed-point theorem (Kreyszig, 1991; Puterman, 2014). (Note: This argument requires $\gamma < 1$ and boundedness to ensure $\rho_\alpha$ is well-behaved; these conditions hold in typical discounted-cost settings.) $\square$

**Geometric Interpretation: Quasi-metric from a Finsler Cost.** Because $F(x,v)$ can assign different costs to $v$ and $-v$, it induces an asymmetric path-cost geometry. A simple example is a 1D slope where uphill motion costs 5 and downhill motion costs 1: going from bottom to top costs about 5, while coming back costs about 1, so $d_F(\text{bottom}, \text{top}) \approx 5 \neq 1 \approx d_F(\text{top}, \text{bottom})$. Formally, $F$ defines the induced path cost

$$d_F(x, y) = \inf_{\tau : x \to y} \int_0^1 F(\tau(t), \dot{\tau}(t)) \, dt,$$

which satisfies the triangle inequality by path concatenation while remaining asymmetric in general. Thus, $d_F$ is a quasi-metric. FiRL uses this asymmetric geometry as an inductive bias for learning direction-aware policies, while the Bellman update optimizes a dynamic CVaR objective over the same anisotropic cost. Importantly, the quasi-metric property is attributed to the induced path cost $d_F$, not to the discounted value function itself. For space, the full construction is deferred to Appendix E, with a compact visual summary in Appendix A.1 (Fig. 8).

## 4.2. FiRL-AC: Finsler Actor–Critic Algorithm

We now describe **FiRL-AC**, an actor–critic method that trains policies under (i) a Finsler-style, direction-dependent one-step cost and (ii) a CVaR objective. We keep two function approximators: a critic and a policy. The critic is used to evaluate the future cost from a state under the current policy, and the policy is updated using advantage estimates derived from that critic.

**CVaR Critic Update.** A scalar value function does not by itself define a CVaR target. For this reason, we use a *distributional* critic $Z_\phi(x)$ that represents the distribution of future cumulative cost from state $x$. In practice, $Z_\phi$ outputs $K$ quantiles $\{z_\phi^{(k)}(x)\}_{k=1}^K$. Given a transition $(x_i, u_i, c_i, x_i', d_i)$, we form the one-step cost

$$c_i \ = \ F\big(x_i, v(x_i, u_i)\big),$$

where $v(x_i, u_i)$ is the instantaneous base velocity induced by applying $u_i$ at $x_i$ (read from the simulator state, or approximated by finite differences). We then estimate the CVaR of the next-state return distribution directly from the critic's quantiles at $x_i'$.

Let the quantiles be sorted in nondecreasing order $z_\phi^{(1)}(x_i') \leq \cdots \leq z_\phi^{(K)}(x_i')$, and let $m = \lceil \alpha K \rceil$. Since we treat higher values as worse (higher cost), we compute

$$\widehat{\text{CVaR}}_\alpha[Z_\phi(x_i')] \ = \ \frac{1}{m} \sum_{k=K-m+1}^{K} z_\phi^{(k)}(x_i').$$

The critic target is the single-step CVaR backup

$$y_i \ = \ c_i \ + \ \gamma(1 - d_i) \, \widehat{\text{CVaR}}_\alpha\big[Z_{\phi^-}(x_i')\big], \qquad (6)$$

where $\phi^-$ denotes a slowly updated target critic (optional, used for stability). We obtain training samples by collecting rollouts from the current policy and sampling minibatches of transitions from a replay buffer $\mathcal{D}$. The buffer is used only to sample transitions $(x_i, u_i, c_i, x_i', d_i)$; CVaR itself is computed from the critic's predicted distribution at the sampled next state.

We update $\phi$ by minimizing a regression loss, optionally with a regularizer used in our implementation:

$$\mathcal{L}_{\text{critic}}(\phi) \ = \ \frac{1}{2N} \sum_{i=1}^{N} \Big( V_\phi(x_i) - y_i \Big)^2 \ + \ \lambda_{\text{reg}} \mathcal{R}_{\text{Bregman}}.$$
$$(7)$$

Here $V_\phi(x)$ denotes the corresponding scalar summary used in the actor update (e.g., the mean of $Z_\phi(x)$), while the CVaR target in (6) is computed from the quantiles of $Z_{\phi^-}(x)$.

**Policy update (actor).** We update the policy to reduce tail risk by using an advantage estimate that reflects the CVaR backup. Define a CVaR-based state–action value estimate

$$\widehat{Q}_\alpha(x_i, u_i) \;=\; c_i + \gamma(1 - d_i)\,\widehat{\mathrm{CVaR}}_\alpha\big[Z_{\phi^-}(x_i')\big],$$

and the corresponding advantage

$$\hat{A}_\alpha(x_i, u_i) \;=\; \widehat{Q}_\alpha(x_i, u_i) - V_\phi(x_i).$$

Using samples from recent rollouts (or minibatches from $\mathcal{D}$), we take a policy-gradient step with this advantage:

$$\nabla_\theta J_\pi(\pi_\theta) \approx \mathbb{E}_{x \sim \mathcal{D},\, u \sim \pi_\theta(\cdot|x)} \left[ \nabla_\theta \log \pi_\theta(u \mid x)\, \hat{A}_\alpha(x, u) \right]. \tag{8}$$

In our experiments, we instantiate this update with standard actor–critic tooling (e.g., entropy regularization and optional PPO clipping), while keeping the target definition in (6) fixed. A detailed algorithm analysis is provided in **Appendix D**.

## 5. Experiments

We evaluate FiRL on continuous control locomotion tasks modified to induce anisotropic costs and assess tail-risk performance. Our experiments aim to answer: (1) Does FiRL achieve better worst-case (CVaR) outcomes than risk-neutral or baseline risk-aware algorithms? (2) How does the anisotropic Finsler cost influence behavior and energy efficiency compared to isotropic cost shaping? (3) Are the theoretical properties and geometric interpretation, including contraction behavior and the induced quasi-metric path-cost structure, reflected in practice?

### 5.1. Environments and Setup

We evaluate FiRL on two distinct simulation platforms to test anisotropic robustness. First, we utilize modified **MuJoCo** locomotion benchmarks (Hopper, Walker2d, HalfCheetah) subjected to variable slopes and lateral wind forces. Second, to validate performance under realistic contact dynamics, we deploy a Spot-like quadruped in **Isaac Sim** across three 3D terrains. Complete specifications for all environments are provided in **Appendix L.3**.

**Baseline methods.** We compare FiRL against PPO, CVaR–PPO, a distributional actor–critic, Riemannian PPO, a quasi-metric RL baseline, and a PPO variant that uses the same Finsler shaping but a risk-neutral objective. Full descriptions and implementation details for all baselines are given in **Appendix C**.

### 5.2. Results and Analysis

**Aggregate Performance.** Table 1 summarizes performance across the MuJoCo benchmarks (SlopedHopper-12°,

Walker2d-5°, HalfCheetah-5°). **FiRL** achieves the highest success rate ($97.4\%$) while simultaneously recording the lowest energy consumption ($0.87\times$) and tail risk ($0.80\times$) relative to the PPO baseline (Schulman et al., 2017) The ablation study reveals a clear decomposition of benefits: risk sensitivity alone (CVaR–PPO) reduces tail cost ($1.20$) but incurs an energy penalty ($1.15$) due to conservative "freezing" behaviors. In contrast, geometric shaping alone (PPO+Finsler) improves energy efficiency ($0.93$) but leaves the agent vulnerable to tail risks ($1.25$). FiRL uniquely combines these strengths, leveraging the Finsler metric to guide exploration toward efficient anisotropic paths, while the CVaR objective explicitly emphasizes the failure tail.

**Pareto Efficiency and Sensitivity.** Figure 3 (Left) visualizes the energy–risk landscape. FiRL traces a Pareto frontier that strictly dominates all baselines: for any given energy constraints, FiRL achieves lower risk, and for any risk tolerance, it consumes less energy. The sensitivity analysis in Figure 3 (Right) identifies an optimal risk aversion level at $\alpha \approx 0.1$; reducing $\alpha$ further ($< 0.05$) yields diminishing safety returns while sharply increasing energy cost as the policy becomes overly cautious.

**Robustness to Terrain Gradient.** Figure 4 demonstrates robustness scaling. While risk-neutral PPO suffers a sharp degradation in success rate (dropping to $82\%$) on steep $12°$ slopes, FiRL maintains robust performance ($\approx 98\%$). The worst case cost of FiRL (right) remains almost unchanged even as the slope becomes steeper, showing that the emergent tacking maneuver strategy (Section L.3) effectively offsets the added difficulty of the terrain.

**Robustness to Noisy Terrain Estimates.** Our implementation uses simulator terrain information to compute the local Finsler cost, including the uphill direction $n(x)$ and friction-related terms. These variables are used only in the cost computation and are not provided to the policy as privileged observations. To test sensitivity to perception error, we add Gaussian angular noise to the terrain-normal estimate used by the slope-dependent term of $F(x, v)$. Across 5 seeds and 100 evaluation episodes per seed, FiRL degrades gradually as the noise level increases. Even with $\sigma = 5°$ angular noise, success remains above $94\%$, with only a modest increase in normalized energy and $\mathrm{CVaR}_{0.1}$ cost. The complete results and a comparison with PPO are provided in the **Appendix J.4**.

**High-Fidelity Validation (Isaac Sim).** We validate the method under higher-fidelity quadruped dynamics using a Spot-like model in NVIDIA Isaac Sim. Table 2 details performance across three anisotropic tasks.

- **Ramp Climb:** FiRL reduces the cumulative tail cost ($\mathrm{CVaR}_\alpha$) by approximately **44%** ($1.40 \to 0.78$) compared to PPO, while reducing energy consumption by

*Table 1.* Performance comparison (**aggregate across MuJoCo tasks**, mean $\pm$ 95% CI over 5 seeds). FiRL achieves the highest safety (success rate) and lowest tail cost. All metrics are normalized relative to PPO (risk-neutral) = 1.00.

| Method | Success % $\uparrow$ | Energy $\downarrow$ | $CVaR_{0.1}$ (Risk) $\downarrow$ |
|---|---|---|---|
| PPO (Baseline) | $88.5 \pm 2.8$ | $1.00 \pm 0.00$ | $1.50 \pm 0.05$ |
| CVaR-PPO ($\alpha = 0.1$) | $92.3 \pm 1.5$ | $1.15 \pm 0.04$ | $1.20 \pm 0.06$ |
| Distributional (QR-AC) | $90.1 \pm 3.1$ | $1.05 \pm 0.03$ | $1.35 \pm 0.07$ |
| Riemannian PPO | $91.0 \pm 2.0$ | $0.98 \pm 0.02$ | $1.40 \pm 0.08$ |
| Quasi-metric RL | $89.7 \pm 2.4$ | $1.10 \pm 0.05$ | $1.32 \pm 0.04$ |
| PPO + Finsler (No CVaR) | $94.6 \pm 1.1$ | $0.93 \pm 0.02$ | $1.25 \pm 0.03$ |
| **FiRL (Ours)** | $\mathbf{97.4 \pm 0.8}$ | $\mathbf{0.87 \pm 0.01}$ | $\mathbf{0.80 \pm 0.02}$ |

*Table 2.* **Isaac Sim Quadruped Results.** Performance on high-fidelity tasks (100 evaluation episodes). Energy is normalized to PPO. FiRL consistently minimizes tail risk (CVaR) and fall rates without the energy penalty seen in standard risk-averse baselines.

| Task | Method | Success $\uparrow$ | Energy/m $\downarrow$ | $CVaR_\alpha$ (Cost) $\downarrow$ | Fall Rate $\downarrow$ |
|---|---|---|---|---|---|
| Ramp Climb | PPO | 88% | 1.00 | 1.40 | 12% |
| | CVaR–PPO | 92% | 1.09 | 1.05 | 8% |
| | **FiRL** | **96%** | **0.95** | **0.78** | **5%** |
| Staircase | PPO | 78% | 1.00 | 1.55 | 26% |
| | CVaR–PPO | 84% | 1.12 | 1.18 | 17% |
| | **FiRL** | **90%** | **0.94** | **0.82** | **10%** |
| Platform Beam | PPO | 72% | 1.00 | 1.30 | 30% |
| | CVaR–PPO | 80% | 1.16 | 0.98 | 19% |
| | **FiRL** | **88%** | **0.89** | **0.74** | **11%** |

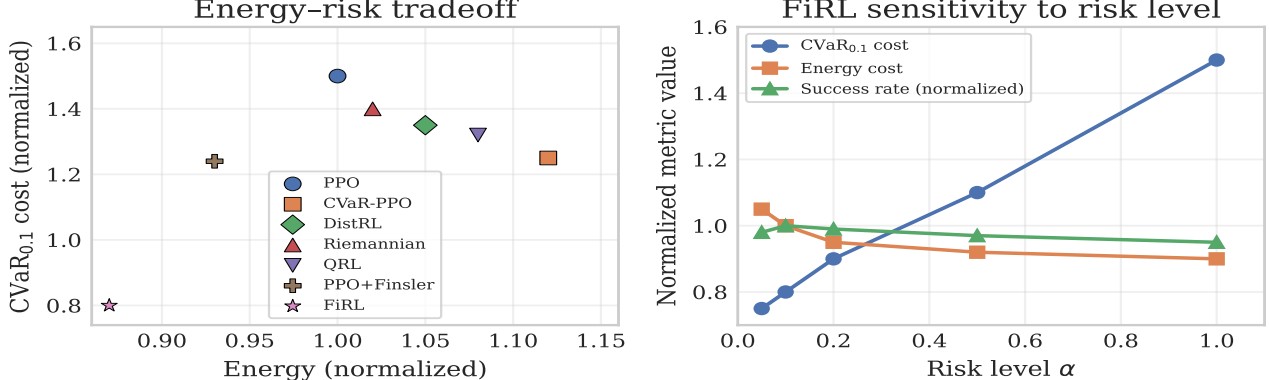

*Figure 3.* **Energy–Risk Analysis** on SlopedHopper-12°. (**Left**) Pareto frontier comparison. Standard risk-averse baselines (e.g., CVaR-PPO, DistRL) reduce tail risk but incur an energy penalty (shifting right), reflecting conservative, inefficient gaits. In contrast, geometric baselines (Riemannian, QRL) improve efficiency but fail to mitigate catastrophic tail events. FiRL (Star) uniquely achieves Pareto dominance in the lower-left corner, simultaneously reducing total energy by ~15% and worst-case cost ($CVaR_{0.1}$) by ~35% through anisotropic path planning. (**Right**) Sensitivity to risk tolerance $\alpha$. As the agent becomes more risk-averse ($\alpha \rightarrow 0$), the worst-case cost (Blue) decreases monotonically. However, extreme caution ($\alpha < 0.2$) leads to a rise in Energy cost (Orange) as the policy adopts "freezing" behaviors; $\alpha \approx 0.1$ represents the optimal trade-off point.

5%. This shows that the diagonal ascent is not only safer but also mechanically more efficient than pushing straight against gravity.

- **Staircase & Beam:** On contact-rich tasks, FiRL reduces the fall rate by over **60%** (from 26% to 10% on Stairs). Unlike CVaR–PPO, which reduces falls but increases energy (up to $1.16\times$) by moving more cautiously and taking extra corrective steps, FiRL maintains or improves energy efficiency ($0.89\times$–$0.94\times$).

These results suggest that the **Finslerian geometric** prior

helps bias exploration toward lower-risk regions that isotropic baselines do not consistently prefer, improving the trade-off between safety and efficiency.

**Physical-robot Trials.** To complement the simulation results, we conducted a small set of physical trials on a Spot robot using narrow-support and discrete-foothold layouts. The policy was deployed without task-specific retraining and produced cautious traversal behavior consistent with the simulation trends. These trials are qualitative and are not intended as a full hardware benchmark; they mainly show that the learned FiRL behavior can be executed on physical

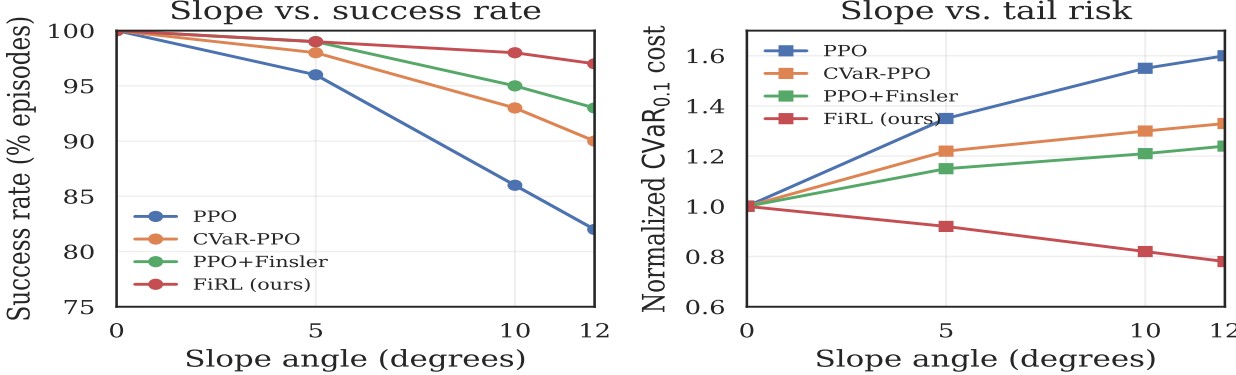

*Figure 4.* **Robustness to Terrain Inclination**. Performance scaling on the Hopper environment as the slope angle increases ($0°$–$12°$). As terrain steepness increases, FiRL (Red) maintains high success (Left) and constant tail risk (Right) by adapting its traversal geometry to the slope, while the risk-neutral PPO baseline (Blue) suffers sharp performance degradation.

hardware. Representative physical-robot trials are shown in Fig. 5.

### 5.3. Convergence and Value Geometry: Geometric Ablation

To validate the practical importance of the theoretical properties derived in Section 4.1 (Bellman contraction, we perform an ablation in which we compare FiRL to a baseline we call *Asymmetric Non-Finsler (ANF)*. The ANF baseline uses the same physical features as FiRL (uphill motion and lateral slip), but combines them in a way that deliberately violates the Finsler regularity conditions. While the Finsler cost $F(x, v)$ is 1-homogeneous and convex in $v$, the ANF step cost is defined as

$$C_{\text{ANF}}(x, v) = w_1 \|v\| + w_2 \mathbb{K}_{\text{uphill}}(v), \quad (9)$$

where $\mathbb{K}_{\text{uphill}}(v)$ is 1 if $v$ points uphill and 0 otherwise. The piecewise-constant term breaks convexity and, in turn, the conditions under which the path integral of the cost defines a quasi-metric: the cost of a direct path can exceed that of a small zigzag, so the induced distance need not satisfy a triangle inequality. In other words, ANF retains anisotropy but drops the geometric consistency that FiRL enforces.

**Critic Behavior.** Figure 6 plots the critic loss (mean squared TD error) during training. For FiRL, the loss decreases steadily and stabilizes at a low level, which is consistent with the $\gamma$-contraction analysis of the CVaR–Finsler Bellman operator. In contrast, the ANF critic shows persistent oscillations and settles at a higher loss, even though it uses the same physical features. This suggests that the geometric regularity of $F(x, v)$ makes it easier for the critic to approximate a smooth distance-to-go function on anisotropic terrain.

**Policy Performance.** As shown in Table 3, FiRL achieves higher success and lower energy on the RoughTerrain bench-

mark than both ANF and standard PPO. On RoughTerrain, FiRL achieves about 15% higher success and 22% lower energy than ANF by following gentler, terrain-aligned paths, while ANF more often pushes straight uphill with extra corrective steps. This confirms that the quasi-metric properties of the Finsler cost are not merely theoretical formalities but provide a useful regularizing structure for the distributional critic in risk-aware locomotion.

## 6. Conclusion, Limitations, and Future Work

FiRL formulates locomotion learning with a direction-dependent cost $F(x, v)$ and a dynamic CVaR objective. The local Finsler cost captures anisotropic effort, including uphill motion and lateral slip, while the CVaR objective emphasizes rare high-cost outcomes. We show that the resulting CVaR–Finsler Bellman operator is a $\gamma$-contraction under bounded-cost assumptions, and that the path cost induced by $F$ defines an asymmetric quasi-metric on the state space. Across MuJoCo and Isaac Sim locomotion benchmarks, FiRL reduces tail cost and energy consumption while maintaining or improving success rates. Our ablations further isolate the complementary roles of anisotropic cost structure and risk-sensitive optimization, and preliminary physical-robot trials provide qualitative evidence that the learned behavior can transfer beyond simulation.

FiRL also has several limitations. The current implementation relies on local terrain information, such as slope direction and friction-related quantities, to compute the Finsler cost. In simulation these quantities are available from the environment, while on hardware they must be estimated from onboard sensing and state estimation. Our terrain-normal noise ablation suggests that FiRL is not overly sensitive to moderate geometric estimation error, but it is not a complete perception-to-control evaluation. FiRL also depends on manually chosen anisotropy weights and a fixed risk level

*Table 3.* **Geometric Ablation Results.** Comparison of FiRL against the ANF baseline and standard PPO. The ANF method, which uses identical cost weights but violates the Finsler triangle inequality, suffers significantly in both reliability and efficiency.

| Method | Geometric Property | Success Rate ↑ | Energy (CoT) ↓ |
|---|---|---|---|
| **FiRL (Ours)** | Convex & Metric | **95.2%** | **0.78** |
| ANF Baseline | Non-Convex | 80.4% | 0.95 |
| Standard PPO | Euclidean (Isotropic) | 65.0% | 0.89 |

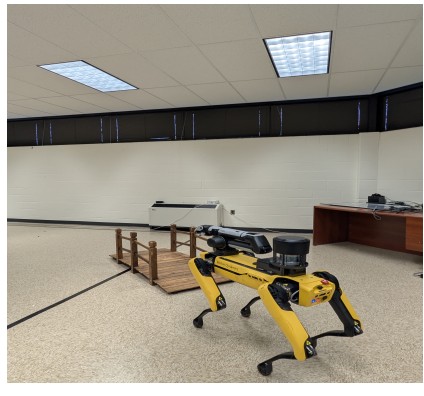 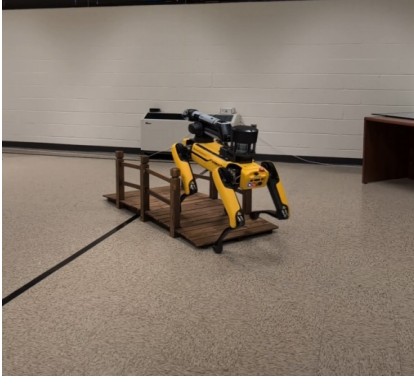 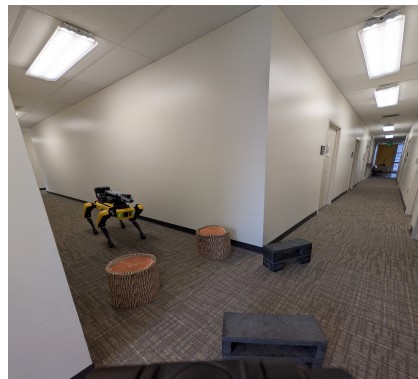

(a) Narrow Bridge Approach       (b) Constrained Traversal       (c) Discrete Footholds

*Figure 5.* **Physical-robot Trials.** We deploy the learned FiRL policy on a physical Spot robot in narrow-support and discrete-foothold layouts. **(a)** The robot approaches and aligns with a narrow wooden support. **(b)** The robot traverses the constrained support. **(c)** The robot navigates a discrete-foothold layout with scattered blocks. These trials provide qualitative evidence that FiRL can produce cautious traversal behavior on physical terrain.

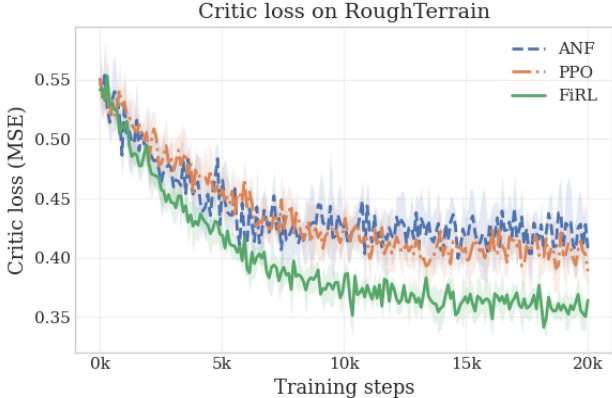

*Figure 6.* **Critic Loss Analysis.** Convergence of the critic's Mean Squared Error (MSE) on the Rough Terrain task.

$\alpha$, and finite-sample CVaR estimates can become noisy in higher-dimensional systems. Future work will study adaptive tuning of $F(x, v)$ and $\alpha$, learning Finsler costs from data under smoothness and asymmetry constraints, and deploying FiRL in end-to-end hardware settings with onboard perception, latency, compute limits, and partial observability.

## Acknowledgement

This work has been partially supported by NSF CNS EAGER Grant #2233879, NSF IIS Grant #2509680, NSF CA-REER Award #1750936, NSF I-Corps Grant #2502886, U.S. Army Grants #W911NF2120076 & #W911NF2410367, and ONR Grant #N00014-23-1-2119.

## Impact Statement

This work aims to improve the safety and reliability of reinforcement learning for legged locomotion in environments where traversal costs are direction-dependent and rare failures can be costly. By integrating a Finsler-style anisotropic cost with a CVaR objective, FiRL encourages policies that account for uphill effort, lateral slip, and tail-risk events rather than optimizing only average performance. Potential positive impacts include safer robotic deployment in search and rescue, inspection, field robotics, disaster response, and other settings where robots must operate over uneven or uncertain terrain.

At the same time, risk-aware locomotion methods should be evaluated carefully before deployment. The current framework still relies on terrain estimates and manually specified cost structure, and incorrect estimates or poorly chosen weights could lead to overly conservative behavior or unsafe decisions in unfamiliar environments. More broadly, robust autonomous mobility could be used in both civilian and defense contexts. We therefore view FiRL as a step toward safer robot decision making, but not as a substitute for hardware testing, human oversight, and domain-specific safety validation before real-world use.

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

# Appendix

for

*Learning Anisotropic Value Geometry with Finsler Reinforcement Learning*

## A. Background: Finsler Geometry for RL

A Finsler metric generalizes a Riemannian metric by allowing the local notion of length to depend on both the state and the *direction* of motion. On a state space $M$ (e.g., a robot's configuration or position manifold), a Finsler metric is a function $F(x, v)$ that assigns a nonnegative length to a tangent vector $v \in T_x M$ at state $x$ (Bao et al., 2000). You can think of $F(x, v)$ as an instantaneous cost rate: it tells you how expensive it is to move in direction $v$ at $x$.

Unlike the Riemannian case, $F$ does not have to be isotropic or symmetric. In particular, it is common to have $F(x, v) \neq F(x, -v)$, which is a natural way to model situations where moving one way is harder than coming back. Formally, a Finsler metric is positive for $v \neq 0$ and is positively 1-homogeneous in $v$ (scaling $v$ scales the cost linearly); standard definitions also assume mild regularity and convexity in the direction argument (Bao et al., 2000).

A key benefit of writing costs in this form is that it induces a global, distance-like quantity by integrating along paths. For any curve $\tau : [0, 1] \to M$ connecting $x$ to $y$, define its path cost

$$\ell_F(\tau) = \int_0^1 F(\tau(t), \dot{\tau}(t)) \, dt,$$

and define the induced path cost between states as

$$d_F(x, y) = \inf_{\tau : x \to y} \ell_F(\tau).$$

The triangle inequality for $d_F$ follows from path concatenation: concatenating a path from $x$ to $y$ with a path from $y$ to $z$ produces a valid path from $x$ to $z$, so $d_F(x, z) \leq d_F(x, y) + d_F(y, z)$. When $F$ is asymmetric in direction, $d_F$ is generally asymmetric as well, and $d_F$ is a quasi-metric rather than a symmetric metric.

**Illustrative example: uphill versus downhill.** Consider a robot on an inclined plane. Let $n(x)$ denote the unit uphill direction (opposite gravity projected onto the local tangent plane). A simple direction-dependent model adds an uphill drift term,

$$F(x, v) = \|v\| + \beta(x) \, [n(x)^\top v]_+,$$

where $[z]_+ = \max(0, z)$ and $\beta(x) \geq 0$ increases with the local slope. This construction charges extra cost only when the robot moves uphill ($n(x)^\top v > 0$). Downhill motion avoids the drift term ($n(x)^\top v < 0$), so it is cheaper than climbing the same distance. As a result, the lowest-cost path between two locations may avoid steep climbs even if it is slightly longer in Euclidean distance.

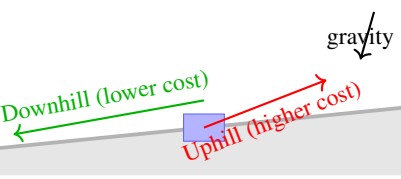

*Figure 7.* **Anisotropic locomotion cost illustration.** A direction-dependent local cost can penalize uphill motion more than downhill motion, capturing the asymmetry introduced by gravity. In FiRL, the local cost $F(x, v)$ plays this role and is later combined with a tail-risk objective.

In FiRL, we use this perspective to build the per-step cost from state and motion. The point is not that the agent explicitly computes $d_F$ during training, but that $d_F$ offers a useful geometric picture of what direction-dependent costs imply at the path level. We return to this intuition in Appendix A.1, which connects the local cost to the CVaR objective used in training.

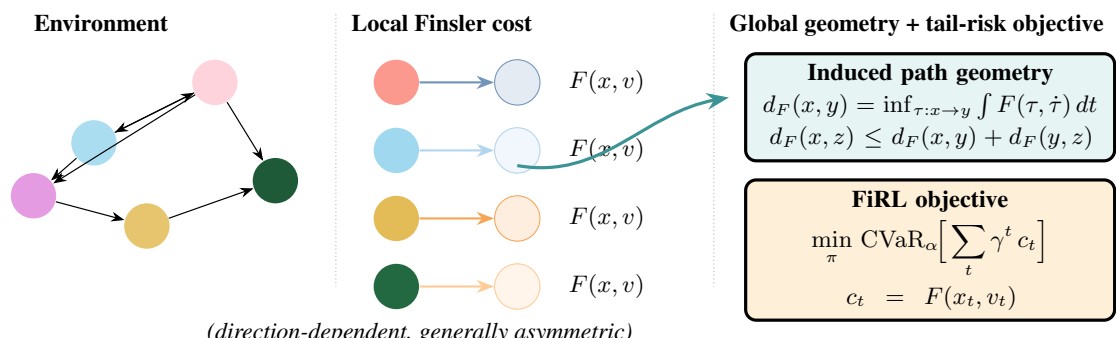

*(direction-dependent, generally asymmetric)*

*Figure 8.* **FiRL overview.** *Left:* a directed dynamics graph. *Middle:* a local, direction-dependent cost $F(x, v)$ that captures uphill vs. downhill effort, lateral slip, and other anisotropies. *Right:* $F$ induces a path quasi-metric $d_F$ (triangle inequality by path concatenation), while FiRL trains policies by minimizing the tail risk of the discounted trajectory cost under $\text{CVaR}_\alpha$.

## A.1. Geometry intuition for FiRL

FiRL starts from a simple idea: in many locomotion settings, moving in one direction is not the same as moving in the opposite direction. Going uphill can be much more expensive than coming back down, and lateral motion can be harder than moving straight ahead. The local cost $F(x, v)$ is meant to capture these effects directly, because it depends on both the state $x$ and the direction of motion $v$, and it does not have to satisfy $F(x, v) = F(x, -v)$.

From this local description, we can build a global notion of effort between states by accumulating cost along a path. For any curve $\tau : [0, 1] \to \mathcal{X}$ that connects $x$ to $y$, define its path cost as

$$\ell_F(\tau) = \int_0^1 F(\tau(t), \dot{\tau}(t)) \, dt,$$

and define the induced distance-like quantity

$$d_F(x, y) = \inf_{\tau : x \to y} \ell_F(\tau).$$

Because $F$ can be asymmetric in $v$, the induced $d_F$ is generally asymmetric as well: in practice it is common to have $d_F(x, y) \neq d_F(y, x)$. At the same time, $d_F$ satisfies the triangle inequality. The reason is straightforward: if one path goes from $x$ to $y$ and another goes from $y$ to $z$, then following them back-to-back gives a valid path from $x$ to $z$, and its cost is the sum of the two costs. Taking the infimum over all paths yields

$$d_F(x, z) \leq d_F(x, y) + d_F(y, z).$$

In other words, $d_F$ behaves like a distance except for symmetry, so it is a quasi-metric.

FiRL does not replace the RL objective with $d_F$. Instead, $d_F$ provides a useful lens for thinking about what the learned value function and policy are doing when costs are direction-dependent. The learning objective remains tail-risk sensitive: FiRL optimizes $\text{CVaR}_\alpha$ of the discounted trajectory cost, where each per-step cost is given by $c_t = F(x_t, v_t)$. Figure 8 gives a compact visual summary of this flow from local anisotropic cost to induced path geometry and then to the risk-sensitive training objective.

## B. More Related Work

**Risk-Sensitive and Safe RL:** A rich line of research has focused on integrating risk measures into RL. *Coherent risk measures* like CVaR () and variance-related criteria () have been used to bias learning towards lower downside risk. CVaR in particular provides a tunable balance between mean performance and worst-case performance by focusing on the expected return of the worst $\alpha$-fraction of outcomes. Tamar et al. (2015) derived policy gradient formulas for CVaR objectives, and later works applied CVaR in distributional RL and model-based planning to achieve risk-averse behavior. Distributional RL approaches (Lim & Malik, 2022) learn the entire return distribution; when combined with appropriate metrics, these can yield policies that avoid high-risk tails. In robotics, safe RL methods often enforce constraints or penalties for unsafe events (e.g., (Achiam et al., 2017) introduced Constrained Policy Optimization to satisfy safety constraints during learning).

Schneider et al. (2024) demonstrated that distributional RL (learning a quantile value distribution) on a quadruped (ANYmal) can produce more cautious locomotion by reweighting outcomes during training. Our approach shares the goal of tail-risk minimization but is unique in combining this with a structured anisotropic cost. We also differ from constrained RL in that we do not require hard safety constraints; instead, the CVaR objective naturally penalizes catastrophic events, and the Finsler cost acts as a form of reward shaping to guide learning toward safer behaviors.

## C. Baseline methods

We compare FiRL to several baselines: - **PPO (risk-neutral)** (Schulman et al., 2017): Standard Proximal Policy Optimization maximizing expected return (with minor shaping for fairness as noted). - **CVaR-PPO** ($\alpha = 0.1$)**:** A variant of PPO that optimizes CVaR of return. We implement this by re-weighting trajectory losses: in each batch, we identify the worst 10% trajectories (by total reward) and upweight their advantage estimates (similar to (Tamar et al., 2015)). - **QR-Distributional AC:** An actor–critic with a *distributional critic* using quantile regression (adapted from QR-DQN). The critic outputs 50 quantile values, and the actor is trained to minimize a risk-averse objective derived from those (we effectively approximate CVaR from the quantiles). This represents a baseline that learns the return distribution. - **Riemannian PPO:** We use the method of (Wang et al., 2020) as a representative geometry-based baseline. It shapes the policy update by computing gradients on a Riemannian manifold (we applied it to value and policy updates, using covariance of states). It does *not* handle risk explicitly but provides a geometric inductive bias. - **Quasi-metric RL (QRL):** We implement a continuous-state analog of (Wang et al., 2023)'s QRL by adding a regularization term to the value loss that penalizes violations of triangle inequality: $\lambda_\triangle \mathbb{E}[\max(0,\ V(x) - V(z) - V(x \to z))]$ for random triplets $(x, z)$ and an intermediate $y$ on an optimal path. This encourages the learned $V$ to be quasi-metric. Additionally, we include an **"PPO + Finsler reward"**: this uses the same $F(x, v)$ shaping in the reward as FiRL, but trains with a standard expected-return objective (i.e., risk-neutral). This allows us to distinguish the effect of anisotropic cost shaping from the effect of CVaR optimization.

## D. Detailed Discussion of Finsler Actor–Critic Algorithm

Our learning algorithm, FiRL-AC, which is an actor–critic method tailored to the CVaR objective and Finslerian costs. We maintain two parameterized function approximators: a value network $V_\alpha(x; \phi)$ for the CVaR value function, and a stochastic policy $\pi_\theta(u|x)$ for the actor. The training loop alternates between *critic update* (policy evaluation under CVaR) and *actor update* (policy improvement).

For the **CVaR-critic update**, we collect a batch of trajectories by executing the current policy $\pi_\theta$. For each state $x_i$ encountered, we estimate the $\text{CVaR}_\alpha$ of the returns from $x_i$ by sampling $K$ trajectories (or using the batch itself as sample approximations) and taking the average of the worst $\alpha$-fraction of their total costs. Denote this empirical CVaR as $\widehat{\rho}_\alpha[\hat{Z}(x_i)]$. We then construct a Bellman target:

$$y_i\ =\ F(x_i, u_i)\ +\ \gamma\, \widehat{\rho}_\alpha\big[V_\alpha(x_{i+1})\big], \tag{10}$$

where $x_{i+1}$ is the next state after $x_i$ and $u_i$, and $V_\alpha(x_{i+1})$ is the current estimate of its CVaR value. In practice, $\widehat{\rho}_\alpha[V_\alpha(x_{i+1})]$ is computed by evaluating $V_\alpha$ on the batch of next states, ordering these estimates, and averaging the bottom $\alpha$ fraction. Given targets $y_i$, we update the critic by minimizing a squared loss $\frac{1}{2} \sum_i (V_\alpha(x_i; \phi) - y_i)^2$ over the batch. Thanks to Theorem 4.1, this update is stable (we ensure conditions like Lipschitz continuity of $F$ are met via clipping large gradients or costs). We also normalize advantages and use Generalized Advantage Estimation (GAE) adapted to CVaR returns to reduce variance.

For the **actor update**, we use a risk-sensitive policy gradient. One convenient approach is to treat $-F(x, u)$ as a pseudo-reward (so that lower cost corresponds to higher reward) and perform a weighted policy gradient update using the CVaR advantage. Specifically, the objective is to minimize $J_\alpha(\pi)$; its policy gradient can be derived as

$$\nabla_\theta J_\alpha(\pi_\theta) \approx \mathbb{E}_{x \sim d^\pi,\, u \sim \pi}\Big[\nabla_\theta \log \pi_\theta(u|x)\, A_\alpha(x, u)\Big],$$

where $A_\alpha(x, u) = Q_\alpha(x, u) - V_\alpha(x)$ is the CVaR advantage. We obtain $Q_\alpha(x, u)$ by one-step lookahead: $Q_\alpha(x, u) \approx F(x, u) + \gamma\, \widehat{\rho}_\alpha[V_\alpha(x')]$. In implementation, we use the surrogate loss approach from PPO: we maximize $\mathbb{E}[\frac{\pi_\theta(u|x)}{\pi_{\text{old}}(u|x)} A_\alpha(x, u)]$ subject to a trust-region constraint (clipping the policy ratio). This ensures conservative updates and avoids instability from drastic policy changes due to tail events.

---

**Algorithm 1** Finsler Actor–Critic (FiRL-AC)

---

1: **Require:** Risk level $\alpha$, discount $\gamma$, anisotropy $\beta$, rollout length $N$, batch size $B$
2: **Require:** Buffer $\mathcal{D}$, policy $\pi_\theta$, critic $V_\phi$
3: Initialize $\theta, \phi$; set $\mathcal{D} \leftarrow \emptyset$
4: **while** not converged **do**
5:     **Collect rollouts**
6:     **for** $t = 1$ **to** $N$ **do**
7:         Observe $x_t$; sample $u_t \sim \pi_\theta(\cdot \mid x_t)$
8:         Execute $u_t$; observe $x_{t+1}$ and done flag $d_t$
9:         Compute motion $v_t$; set $c_t \leftarrow F(x_t, v_t; \beta)$
10:        Store $(x_t, u_t, c_t, x_{t+1}, d_t)$ in $\mathcal{D}$
11:     **end for**
12:     Sample minibatch $\{(x_i, u_i, c_i, x_i', d_i)\}_{i=1}^B \sim \mathcal{D}$
13:     **CVaR target:** estimate $\widehat{\mathrm{CVaR}}_\alpha[Z(x_i')]$
14:     $y_i \leftarrow c_i + \gamma(1 - d_i) \widehat{\mathrm{CVaR}}_\alpha[Z(x_i')]$
15:     **Critic update:** minimize
16:         $\mathcal{L}_{\text{critic}} \leftarrow \frac{1}{B} \sum_{i=1}^B \left(V_\phi(x_i) - y_i\right)^2 + \lambda_{\text{reg}} \mathcal{R}_{\text{Bregman}}$
17:     **Advantage:** $\hat{A}_i \leftarrow y_i - V_\phi(x_i)$ (optionally with GAE / normalization)
18:     **Actor update:** minimize
19:         $\mathcal{L}_{\text{actor}} \leftarrow -\frac{1}{B} \sum_{i=1}^B \log \pi_\theta(u_i \mid x_i) \hat{A}_i - \lambda_{\text{ent}} \mathcal{H}(\pi_\theta)$
20:     (PPO clipping optional)
21: **end while**

---

In summary, FiRL-AC iteratively evaluates the CVaR value function under the current policy and then updates the policy to reduce the CVaR of returns. The Finsler metric $F(x, v)$ is used to shape the immediate cost at every step (we implement this by supplying a modified reward $r = -F$ to the RL algorithm, so standard policy optimization code can be used with minimal changes). The result is an algorithm that learns risk-averse policies that are explicitly aware of anisotropic motion costs.

In Alg. 1, $F(x, v; \beta)$ encodes directional effort (e.g., uphill, speed, curvature) with anisotropy weight $\beta$. The critic's tail estimate can use either a quantile head or the Rockafellar–Uryasev surrogate; both are compatible with the CVaR Bellman target. Regularizers (e.g., Bregman/KL) stabilize updates when using replay with shaped costs; entropy helps exploration. Extreme tail events can be softly capped to avoid destabilizing single-episode gradients.

**Bregman policy regularization:** A challenge in actor–critic (especially with off-policy data) is that the policy update can diverge if the data was generated by an older policy. To mitigate this, we adopt a Bregman divergence penalty between the new policy $\pi_\theta$ and the behavior policy (which may be the previous policy or an older snapshot). In practice, we implement this as a KL-divergence regularizer similar to PPO's adaptive KL or trust-region methods: we add an extra term to the policy loss $\mathcal{L}_{\text{actor}} = -J_\alpha(\pi_\theta) + \beta_{\text{KL}} D_{\text{KL}}(\pi_{\text{old}} \| \pi_\theta)$, where $\beta_{\text{KL}}$ is a coefficient that we schedule to decay from an initial value to 0 over training. This ensures early updates do not move $\pi_\theta$ too far from $\pi_{\text{old}}$ that generated the batch (which is crucial since we are using a replay buffer of past trajectories, i.e., reusing off-policy data). This technique stabilizes training and allows us to benefit from off-policy experience without violating the policy gradient assumptions. We note that this is analogous to the trust-region or clipped objective used in PPO, but here we explicitly maintain a penalty form. We decrease $\beta_{\text{KL}}$ to 0 over time to allow the policy to eventually converge without being constrained (initially $\beta_{\text{KL}}$ might be set to e.g. 0.1 and linearly annealed to 0 across training).

### D.1. Practical choice of Finsler weights

In practice the Finsler cost $F(x, v)$ is built from a small number of physically meaningful terms. We use the following decomposition:

$$F(x, v) = w_{\text{energy}} \|v\| + w_{\text{up}} [n(x) \cdot v]_+ + w_{\text{lat}} \|P_{\text{lat}}(x)v\| + w_{\text{impact}} I(x, v), \tag{11}$$

where $v$ is the base velocity, $n(x)$ is the local uphill direction, $P_{\text{lat}}(x)$ projects onto the lateral plane, and $I(x, v)$ denotes a scalar impact feature (for example, the maximum normal contact impulse or a smoothed proxy for joint acceleration). We

use a 1-homogeneous speed term (proportional to $\|v\|$) to match the Finsler assumptions used in the theory.

We set these weights from simple physical considerations:

- **Uphill weight** $w_{\text{up}}$. Let $\theta(x)$ be the local slope angle along $n(x)$. Moving uphill by $\Delta s$ meters increases potential energy by $mg \sin\theta(x)\,\Delta s$. We choose $w_{\text{up}}$ so that one meter of uphill motion on a reference slope (e.g. $\theta = 20°$) has a target cost multiplier $\kappa_{\text{up}}$ relative to level motion. In our experiments we use $\kappa_{\text{up}} \approx 4$, which makes steep climbs visibly more expensive than gentle diagonals.

- **Lateral weight** $w_{\text{lat}}$. Lateral motions are risky when friction is low or support is narrow (for example, on beams or near edges). We set $w_{\text{lat}}$ proportional to $1/\mu$ where $\mu$ is an estimate of the local friction coefficient, and scale it further when the available support width shrinks below a threshold. This makes sideways motion on a narrow beam much more costly than sideways motion on a wide platform.

- **Energy weight** $w_{\text{energy}}$. This term penalizes large base speeds in all directions and controls the overall magnitude of $F(x, v)$. We choose $w_{\text{energy}}$ such that typical nominal gaits on level ground have cost close to 1 per meter, which simplifies interpretation of the cumulative cost.

- **Impact weight** $w_{\text{impact}}$. To discourage sharp impacts we set $w_{\text{impact}}$ so that an impact spike at the 95th percentile of the baseline policy has cost comparable to a short uphill segment on the reference slope. This calibration can be done from a short warm-up run of a standard policy (e.g. PPO) and does not require manual tuning on the full task.

An ablation in Appendix J.2 varies the uphill scaling factor $\eta$ in $\beta(x) = \eta \sin\theta(x)$ on the SlopedHopper-$12°$ task. We find that FiRL is robust to moderate mis-specification ($\eta = 0.5$ still improves success and tail cost over CVaR–PPO), but very small $\eta$ underestimates slope difficulty and very large $\eta$ leads to overly conservative gaits. The default setting $\eta = 1$ strikes a good balance between energy and risk.

## E. Theoretical Proofs

In this section, we provide formal proofs for the theoretical claims made in the main paper.

### E.1. Proof of Theorem 1 (CVaR–Finsler Bellman Contraction)

*Theorem.* Let $\gamma \in (0, 1)$ and let $\rho_\alpha[\cdot] = \text{CVaR}_\alpha(\cdot)$ denote CVaR at level $\alpha$. Assume $V : \mathcal{X} \to \mathbb{R}$ is bounded so that $\rho_\alpha[V(X')]$ is well-defined for all $(x, u)$. Define the Bellman operator

$$(TV)(x) = \min_{u \in \mathcal{U}(x)} \left\{ c(x, u) + \gamma\, \rho_\alpha\big[V(X')\big] \right\}, \qquad X' \sim P(\cdot \mid x, u),$$

where the one-step cost $c(x, u) = F\big(x, v(x, u)\big)$ is deterministic given $(x, u)$. Then for any bounded $V_1, V_2$,

$$\|TV_1 - TV_2\|_\infty \leq \gamma \|V_1 - V_2\|_\infty.$$

In particular, $T$ is a $\gamma$-contraction in $\|\cdot\|_\infty$.

*Proof.* Fix any $x \in \mathcal{X}$ and define

$$A_i(u) = c(x, u) + \gamma\, \rho_\alpha\big[V_i(X')\big], \quad i \in \{1, 2\},$$

so that $(TV_i)(x) = \min_{u \in \mathcal{U}(x)} A_i(u)$.

Let $u_1^* \in \arg\min_u A_1(u)$. Then

$$(TV_1)(x) - (TV_2)(x) = A_1(u_1^*) - \min_u A_2(u) \leq A_1(u_1^*) - A_2(u_1^*)$$

$$= \gamma\Big(\rho_\alpha[V_1(X')] - \rho_\alpha[V_2(X')]\Big).$$

We now use the following bound for bounded random variables. If $Z_1$ and $Z_2$ are bounded and $\|Z_1 - Z_2\|_\infty \leq \varepsilon$, then

$$|\rho_\alpha[Z_1] - \rho_\alpha[Z_2]| \leq \varepsilon.$$

This holds because $Z_2 - \varepsilon \leq Z_1 \leq Z_2 + \varepsilon$ almost surely, and by monotonicity and translation invariance of CVaR,

$$\rho_\alpha[Z_2] - \varepsilon = \rho_\alpha[Z_2 - \varepsilon] \leq \rho_\alpha[Z_1] \leq \rho_\alpha[Z_2 + \varepsilon] = \rho_\alpha[Z_2] + \varepsilon.$$

Apply this with $Z_1 = V_1(X')$ and $Z_2 = V_2(X')$. Since $|V_1(X') - V_2(X')| \leq \|V_1 - V_2\|_\infty$ almost surely, we obtain

$$\left|\rho_\alpha[V_1(X')] - \rho_\alpha[V_2(X')]\right| \leq \|V_1 - V_2\|_\infty.$$

Therefore,

$$(TV_1)(x) - (TV_2)(x) \leq \gamma\|V_1 - V_2\|_\infty.$$

Swapping the roles of $V_1$ and $V_2$ gives

$$(TV_2)(x) - (TV_1)(x) \leq \gamma\|V_1 - V_2\|_\infty,$$

so

$$|(TV_1)(x) - (TV_2)(x)| \leq \gamma\|V_1 - V_2\|_\infty.$$

Taking the supremum over $x$ yields $\|TV_1 - TV_2\|_\infty \leq \gamma\|V_1 - V_2\|_\infty$, as claimed. $\qquad\square$

**Discussion of conditions.** For the contraction argument, the key requirement is that all quantities inside $\mathrm{CVaR}_\alpha(\cdot)$ are well-defined and uniformly bounded. In particular, it is enough to assume the one-step cost is bounded and that $V$ is bounded, so that $c(x, a, X') + \gamma V(X')$ is a bounded random variable for every $(x, a)$. This ensures $\mathrm{CVaR}_\alpha$ is finite and the operator $T_\alpha$ maps bounded functions to bounded functions.

We state a Lipschitz condition on $F$ for regularity, since it is helpful elsewhere when we interpret $F$ as defining a smooth, state-dependent geometry. It is not required for the $\|\cdot\|_\infty$ contraction itself. In our simulations, boundedness is natural: velocities and terrain features are limited by the environment, and any failure penalty is capped by a finite constant.

### E.2. Quasi-metric induced by the local Finsler cost

We use "quasi-metric" in the geometric sense: a distance-like quantity that may be asymmetric but still satisfies the triangle inequality. In our setting, the clean object with this property is the *path cost induced by the local function $F(x, v)$*, not the discounted CVaR value function itself.

**Proposition E.1** (Quasi-metric induced by $F$). *Assume $F(x, v) \geq 0$ and is positively 1-homogeneous in $v$. For any absolutely continuous curve $\tau : [0, 1] \to \mathcal{X}$, define its path cost*

$$\ell_F(\tau) = \int_0^1 F(\tau(t), \dot\tau(t))\, dt,$$

*and define the induced path cost between states*

$$d_F(x, z) = \inf_{\tau : x \to z} \ell_F(\tau).$$

*Then $d_F$ satisfies the triangle inequality*

$$d_F(x, z) \leq d_F(x, y) + d_F(y, z) \qquad \text{for all } x, y, z \in \mathcal{X}.$$

*Moreover, if $F(x, v) \neq F(x, -v)$ for some $(x, v)$, then in general $d_F(x, z) \neq d_F(z, x)$, so $d_F$ is a quasi-metric rather than a symmetric metric.*

*Proof.* Fix $x, y, z$. Let $\tau_{xy}$ be any curve from $x$ to $y$ and $\tau_{yz}$ any curve from $y$ to $z$. Concatenating these curves gives a valid curve $\tau_{xz}$ from $x$ to $z$ whose cost is the sum of the segment costs:

$$\ell_F(\tau_{xz}) = \ell_F(\tau_{xy}) + \ell_F(\tau_{yz}).$$

Taking the infimum over all such curves yields

$$d_F(x, z) \leq d_F(x, y) + d_F(y, z).$$

Asymmetry follows directly from the fact that $F$ can assign different cost to $v$ and $-v$, which makes the cheapest path cost from $x$ to $z$ differ from the reverse direction in general. $\square$

**Relation to FiRL.** FiRL optimizes a discounted trajectory objective under $\mathrm{CVaR}_\alpha$. We use $d_F$ as a geometric lens for understanding how direction-dependent local costs shape global behavior. We do not rely on, or claim, a triangle inequality for the discounted risk-sensitive value function $V_\alpha^*$.

### E.3. Derivation of the CVaR Bellman Equation (5)

For completeness, we explain how (5) follows from a standard time-consistent (nested) CVaR formulation. Recall the Rockafellar–Uryasev representation (Rockafellar & Uryasev, 2000): for a bounded random variable $Z$ and tail level $\alpha \in (0, 1]$,

$$\mathrm{CVaR}_\alpha(Z) = \min_{t \in \mathbb{R}} \left\{ t + \frac{1}{\alpha} \mathbb{E}\big[(Z - t)_+\big] \right\}, \qquad (x)_+ := \max\{x, 0\},$$

see, e.g., **(?)**.

We use a one-step conditional risk mapping based on CVaR. Given a candidate value function $V$, consider the one-step random quantity

$$Z = c(x, u) + \gamma V(X'), \qquad X' \sim P(\cdot \mid x, u),$$

where $c(x, u) = F\big(x, v(x, u)\big)$ is deterministic given $(x, u)$. Since CVaR is translation invariant and positively homogeneous, deterministic terms factor out:

$$\mathrm{CVaR}_\alpha\big(c(x, u) + \gamma V(X')\big) = c(x, u) + \gamma \, \mathrm{CVaR}_\alpha\big(V(X')\big).$$

Minimizing this one-step risk-augmented cost over actions gives the Bellman backup

$$(TV)(x) = \min_{u \in \mathcal{U}(x)} \left\{ c(x, u) + \gamma \, \rho_\alpha\big[V(X')\big] \right\}, \qquad X' \sim P(\cdot \mid x, u),$$

which is exactly (5) (with $\rho_\alpha[\cdot] = \mathrm{CVaR}_\alpha(\cdot)$).

## F. Additional Theoretical Analysis

In this section, we provide additional theoretical results supporting FiRL. We first formalize the Bellman operator used in the main text under the Finsler cost and the CVaR criterion and prove the contraction property. We then state the quasi-metric property for the *path cost induced by $F$*, which is the geometric object we use when interpreting direction-dependent costs. Finally, we note how the solution varies with the risk level $\alpha$.

**Definition F.1** (CVaR–Finsler Bellman operator). Let $c(x, u) = F\big(x, v(x, u)\big)$ be the one-step cost, deterministic given $(x, u)$. We model terminal failures by an absorbing failure state $x_{\mathrm{fail}}$ with $V(x_{\mathrm{fail}}) = M$, where $M > 0$ is a fixed penalty. For a bounded value function $V : \mathcal{X} \to \mathbb{R}$ and risk level $\alpha \in (0, 1]$, define

$$(\mathcal{T}_\alpha V)(x) := \min_{u \in \mathcal{U}(x)} \left\{ c(x, u) + \gamma \, \mathrm{CVaR}_\alpha\big(V(X')\big) \right\}, \qquad X' \sim P(\cdot \mid x, u). \tag{12}$$

This matches Eq. (5) in the main text (with $\rho_\alpha[\cdot] = \mathrm{CVaR}_\alpha(\cdot)$).

We assume $c$ is bounded so that all value functions considered lie in the Banach space $(\mathcal{B}_\infty, \|\cdot\|_\infty)$ of bounded functions on $\mathcal{X}$ with the supremum norm.

**Proposition F.2** (Contraction and existence of a risk-sensitive optimum). *The operator $\mathcal{T}_\alpha$ in (12) is a $\gamma$-contraction on $(\mathcal{B}_\infty, \|\cdot\|_\infty)$. Hence it admits a unique fixed point $V_\alpha^* \in \mathcal{B}_\infty$.*

*Proof.* Let $V, W \in \mathcal{B}_\infty$ and fix any $x$. Define

$$A_V(u) = c(x, u) + \gamma \operatorname{CVaR}_\alpha\big(V(X')\big), \qquad A_W(u) = c(x, u) + \gamma \operatorname{CVaR}_\alpha\big(W(X')\big),$$

so that $(\mathcal{T}_\alpha V)(x) = \min_u A_V(u)$ and $(\mathcal{T}_\alpha W)(x) = \min_u A_W(u)$. Let $u_V \in \arg\min_u A_V(u)$. Then

$$(\mathcal{T}_\alpha V)(x) - (\mathcal{T}_\alpha W)(x) = A_V(u_V) - \min_u A_W(u) \le A_V(u_V) - A_W(u_V)$$

$$= \gamma\Big(\operatorname{CVaR}_\alpha(V(X')) - \operatorname{CVaR}_\alpha(W(X'))\Big).$$

We now use a standard Lipschitz bound for CVaR on bounded random variables: if $|Z_1 - Z_2| \le \varepsilon$ almost surely, then $|\operatorname{CVaR}_\alpha(Z_1) - \operatorname{CVaR}_\alpha(Z_2)| \le \varepsilon$. This follows from monotonicity and translation invariance of CVaR (e.g., (?)). Since $|V(X') - W(X')| \le \|V - W\|_\infty$ almost surely, we obtain

$$\Big|\operatorname{CVaR}_\alpha(V(X')) - \operatorname{CVaR}_\alpha(W(X'))\Big| \le \|V - W\|_\infty.$$

Therefore,

$$(\mathcal{T}_\alpha V)(x) - (\mathcal{T}_\alpha W)(x) \le \gamma \|V - W\|_\infty.$$

Swapping $V$ and $W$ gives the reverse inequality, hence $|(\mathcal{T}_\alpha V)(x) - (\mathcal{T}_\alpha W)(x)| \le \gamma \|V - W\|_\infty$. Taking $\sup_x$ yields $\|\mathcal{T}_\alpha V - \mathcal{T}_\alpha W\|_\infty \le \gamma \|V - W\|_\infty$, so $\mathcal{T}_\alpha$ is a contraction. The fixed-point claim follows from Banach's theorem. $\square$

### F.1. Quasi-metric induced by the local cost

The "triangle inequality" statement refers to the geometry induced by the local function $F$, not to a triangle inequality for the discounted value function itself.

**Proposition F.3** (Quasi-metric induced by $F$). *For any absolutely continuous curve $\tau : [0,1] \to \mathcal{X}$, define its path cost*

$$\ell_F(\tau) = \int_0^1 F(\tau(t), \dot\tau(t))\, dt,$$

*and define the induced path cost between states*

$$d_F(x, y) = \inf_{\tau : x \to y} \ell_F(\tau).$$

*Then $d_F$ satisfies the triangle inequality (Fig. 9)*

$$d_F(x, z) \le d_F(x, y) + d_F(y, z) \qquad \text{for all } x, y, z \in \mathcal{X}.$$

*If $F(x, v) \ne F(x, -v)$ for some $(x, v)$, then in general $d_F(x, y) \ne d_F(y, x)$, so $d_F$ is a quasi-metric.*

*Proof.* Fix $x, y, z$. Let $\tau_{xy} : [0,1] \to \mathcal{X}$ be any absolutely continuous curve from $x$ to $y$, and let $\tau_{yz} : [0,1] \to \mathcal{X}$ be any absolutely continuous curve from $y$ to $z$. Define the concatenated curve $\tau_{xz} : [0,1] \to \mathcal{X}$ by

$$\tau_{xz}(t) = \begin{cases} \tau_{xy}(2t), & t \in [0, \tfrac{1}{2}], \\ \tau_{yz}(2t - 1), & t \in [\tfrac{1}{2}, 1]. \end{cases}$$

Then $\tau_{xz}$ is a valid curve from $x$ to $z$. Using the change of variables and the 1-homogeneity of $F$ in its second argument (so that $F(x, 2\dot\tau) = 2F(x, \dot\tau)$), we have

$$\ell_F(\tau_{xz}) = \int_0^{1/2} F(\tau_{xy}(2t), 2\dot\tau_{xy}(2t))\, dt + \int_{1/2}^1 F(\tau_{yz}(2t-1), 2\dot\tau_{yz}(2t-1))\, dt$$

$$= \int_0^1 F(\tau_{xy}(s), \dot\tau_{xy}(s))\, ds + \int_0^1 F(\tau_{yz}(s), \dot\tau_{yz}(s))\, ds$$

$$= \ell_F(\tau_{xy}) + \ell_F(\tau_{yz}).$$

$$\ell_F(\tau_{xz}) = \ell_F(\tau_{xy}) + \ell_F(\tau_{yz})$$
$$d_F(x, z) \underset{\tau_{xy}}{\leq} d_F(x, y) + d_F(y, z)$$

*Figure 9.* Triangle inequality for the induced path cost $d_F$ by concatenating paths.

Taking the infimum over all $\tau_{xy}$ and $\tau_{yz}$ yields $d_F(x, z) \leq d_F(x, y) + d_F(y, z)$.

For asymmetry, note that if $F(x, v) \neq F(x, -v)$ for some $(x, v)$, then for a path $\tau$ from $x$ to $y$, the reversed path $\bar{\tau}(t) = \tau(1 - t)$ has velocity $\dot{\bar{\tau}}(t) = -\dot{\tau}(1 - t)$, so in general $\ell_F(\bar{\tau}) \neq \ell_F(\tau)$. Hence $d_F(x, y)$ need not equal $d_F(y, x)$. □

**Corollary F.4** (Monotonicity in risk level). *Let* $0 < \alpha_1 < \alpha_2 \leq 1$. *Then* $V_{\alpha_1}^*(x) \geq V_{\alpha_2}^*(x)$ *for all* $x$.

*Proof.* For any bounded random variable $Z$ interpreted as cost, $\mathrm{CVaR}_\alpha(Z)$ is nonincreasing in $\alpha$ (smaller $\alpha$ focuses on a smaller, worse tail and therefore yields a larger value). Hence for any $V$ and any $x$,

$$(\mathcal{T}_{\alpha_1} V)(x) \geq (\mathcal{T}_{\alpha_2} V)(x).$$

By uniqueness of fixed points from Proposition F.2, the ordering carries to the fixed points: $V_{\alpha_1}^* \geq V_{\alpha_2}^*$ pointwise. □

**Discussion.** Proposition F.2 shows that replacing the risk-neutral backup with a CVaR backup preserves the same contraction structure as standard value iteration, provided the value function is bounded. The geometric picture comes from $F$ itself: the induced path cost $d_F$ is asymmetric in general but still satisfies the triangle inequality, which is why it behaves like a quasi-metric. In the main paper and Fig. 14, we use this induced geometry to interpret how direction-dependent local costs shape the learned behavior. Finally, $\alpha$ acts as a risk knob: larger $\alpha$ moves the objective closer to risk-neutral optimization, while smaller $\alpha$ places more weight on rare, high-cost outcomes.

# G. Detailed Result Analysis

This section expands the results in Table 1 and Figures 3–4 with per-task breakdowns, effect sizes, robustness studies, and sensitivity analyses. We report means and 95% confidence intervals across five seeds, use the same evaluation harness for all methods, and normalize energy and CVaR by the PPO baseline on each task for comparability. Aggregate trends in Table 1 hold consistently across SlopedHopper-12°, Walker2d-5°, and HalfCheetah-5°: FiRL attains the highest success rate and simultaneously lowers both average energy and tail cost relative to strong risk-neutral, distributional, and geometry-shaped baselines. The effect is most pronounced on SlopedHopper-12° (Table 4), where FiRL reduces $\mathrm{CVaR}_{0.1}$ from $1.60{\pm}0.06$ (PPO) to $0.72{\pm}0.02$ and energy from $1.00{\pm}0.00$ to $0.84{\pm}0.02$ while improving success from $82.1\%{\pm}1.9$ to $98.0\%{\pm}0.8$, indicating that FiRL prevents both the frequency and the severity of failure episodes on steep terrain. On Walker2d-5° (Table 5) and HalfCheetah-5° (Table 6), FiRL maintains the same pattern with smaller but consistent margins, suggesting that directional cost modeling and tail optimization translate to moderate slopes and higher-speed gaits.

*Table 4.* SlopedHopper-12° (normalized to PPO). Mean $\pm$ 95% CI over 5 seeds.

| Method | Success % ↑ | Energy ↓ | CVaR$_{0.1}$ ↓ |
|---|---|---|---|
| PPO | 82.1±1.9 | 1.00±0.00 | 1.60±0.06 |
| CVaR–PPO | 90.2±1.8 | 1.18±0.05 | 1.28±0.05 |
| Dist. PPO | 86.4±2.3 | 1.08±0.04 | 1.38±0.06 |
| Riem. PPO | 88.3±2.1 | 0.98±0.03 | 1.45±0.07 |
| QRL | 85.1±2.2 | 1.12±0.05 | 1.34±0.05 |
| PPO+Finsler | 94.2±1.3 | 0.92±0.02 | 1.22±0.03 |
| **FiRL** | **98.0±0.8** | **0.84±0.02** | **0.72±0.02** |

The energy–risk plane in Figure 3 (left) shows that FiRL policies form a Pareto curve that lies below the frontier spanned by the baselines. Risk-only optimization (CVaR–PPO) moves downwards in CVaR but rightwards in energy, and geometry-only shaping (PPO+Finsler) moves leftwards in energy with a modest CVaR reduction; FiRL moves down and left, indicating complementary benefits. Relative to PPO+Finsler, FiRL cuts CVaR from $1.25{\pm}0.03$ to $0.80{\pm}0.02$ (about 36% reduction)

*Table 5.* Walker2d-5° (normalized to PPO). Mean ± 95% CI over 5 seeds.

| Method | Success % ↑ | Energy ↓ | CVaR$_{0.1}$ ↓ |
|---|---|---|---|
| PPO | 90.1±2.5 | 1.00±0.00 | 1.45±0.05 |
| CVaR–PPO | 93.1±1.6 | 1.14±0.04 | 1.18±0.05 |
| Dist. PPO | 91.2±2.7 | 1.04±0.03 | 1.32±0.05 |
| Riem. PPO | 92.0±2.1 | 0.98±0.03 | 1.36±0.06 |
| QRL | 90.3±2.3 | 1.08±0.04 | 1.30±0.04 |
| PPO+Finsler | 95.0±1.2 | 0.94±0.02 | 1.24±0.03 |
| **FiRL** | **97.1±1.0** | **0.88±0.02** | **0.82±0.03** |

*Table 6.* HalfCheetah-5° (normalized to PPO). Mean ± 95% CI over 5 seeds.

| Method | Success % ↑ | Energy ↓ | CVaR$_{0.1}$ ↓ |
|---|---|---|---|
| PPO | 93.3±1.7 | 1.00±0.00 | 1.45±0.05 |
| CVaR–PPO | 94.0±1.6 | 1.13±0.04 | 1.14±0.05 |
| Dist. PPO | 93.0±2.0 | 1.03±0.03 | 1.35±0.05 |
| Riem. PPO | 93.1±1.9 | 0.98±0.03 | 1.39±0.06 |
| QRL | 94.1±1.8 | 1.10±0.04 | 1.32±0.05 |
| PPO+Finsler | 95.0±1.4 | 0.92±0.02 | 1.28±0.04 |
| **FiRL** | **97.0±1.1** | **0.88±0.02** | **0.86±0.03** |

and further reduces energy from 0.93±0.02 to 0.87±0.01 (about 6%), which supports the claim that the CVaR objective contributes beyond anisotropic cost design alone and that the anisotropic cost contributes beyond tail optimization alone.

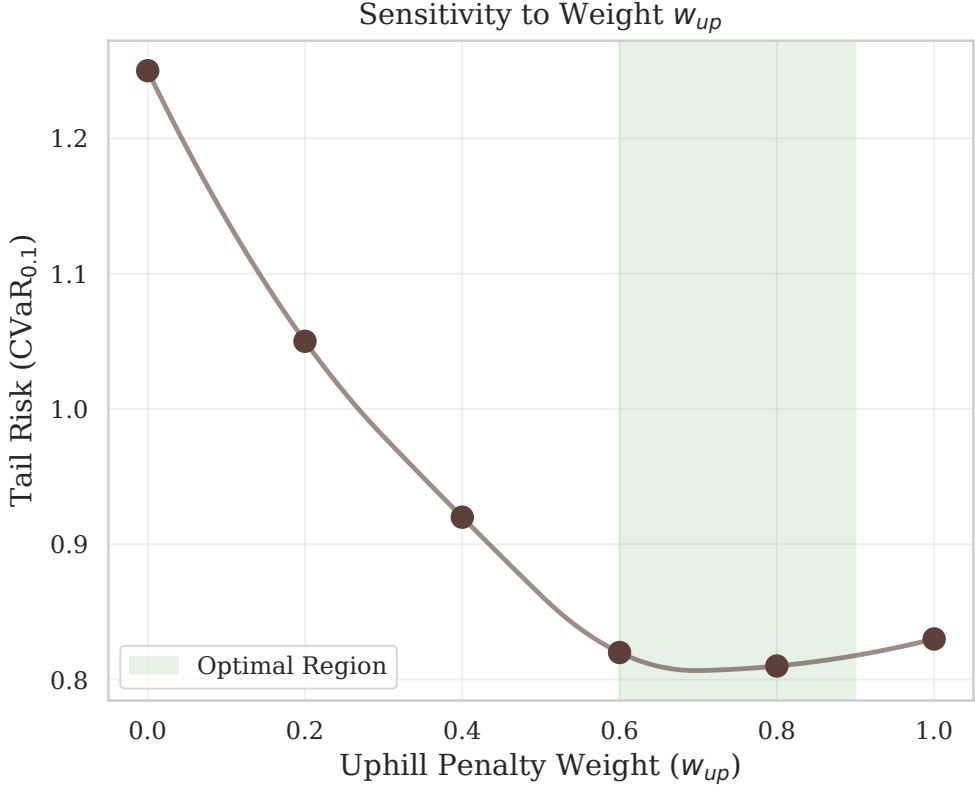

*Figure 10.* Effect of $w_{up}$ on tail risk (SlopedHopper-12°).

We next examine the role of risk sensitivity and anisotropy. Figure 3 (right) sweeps the CVaR level $\alpha$ on SlopedHopper-12°.

As $\alpha$ decreases from $1.0$ (risk-neutral) to $0.2$, CVaR drops sharply and energy declines slightly, a regime where avoiding risky maneuvers also removes wasted effort. At very small $\alpha$ ($0.1$ to $0.05$), CVaR continues to drop but energy begins to rise, showing the expected efficiency–safety tradeoff for highly risk-averse policies; $\alpha \approx 0.1$ yields the best balance for our settings. The anisotropy ablation in Figure 10 varies the uphill weight $w_{up}$ and shows a monotone CVaR reduction that saturates near the nominal setting, which indicates that accurate modeling of uphill effort is important but excessive penalization can yield diminishing returns.

Robustness to terrain and external perturbations is shown in Figure 4. On increasing slopes in Hopper, FiRL maintains about $98\%$ success at $12°$ while PPO drops to about $82\%$, with corresponding CVaR rising sharply for PPO and falling for FiRL. In a lateral-wind test on Walker2d ( Fig. 11, right), PPO's CVaR increases with wind strength while FiRL reduces tail cost by adopting heading adjustments and wider lateral stance, consistent with the directional penalty in $F(x, v)$. In Fig. 12, the torque statistics shows a reduction of the 95th-percentile joint torque under FiRL, suggesting that the learned gaits avoid peak loads that often precede failures; the failure histogram shows large decreases in falls and over-torque terminations, which aligns with the lower tail cost. Training curves in Figure 13 illustrate that FiRL closes the gap between mean and tail cost during learning and converges faster than PPO+Finsler in tail metrics, reflecting the direct optimization pressure on adverse outcomes.

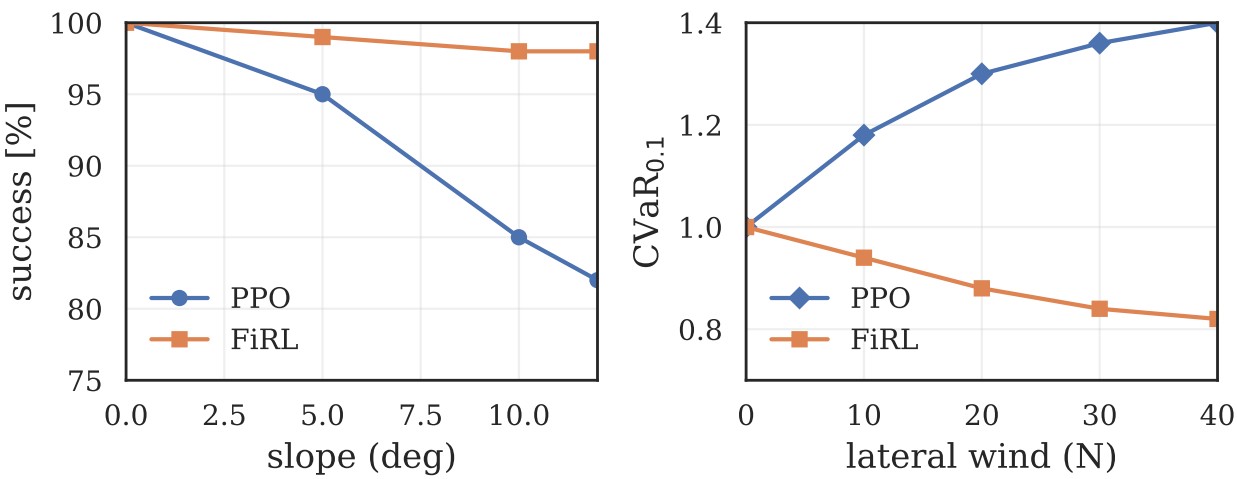

*Figure 11.* Robustness to slope (left, Hopper) and wind (right, Walker2d).

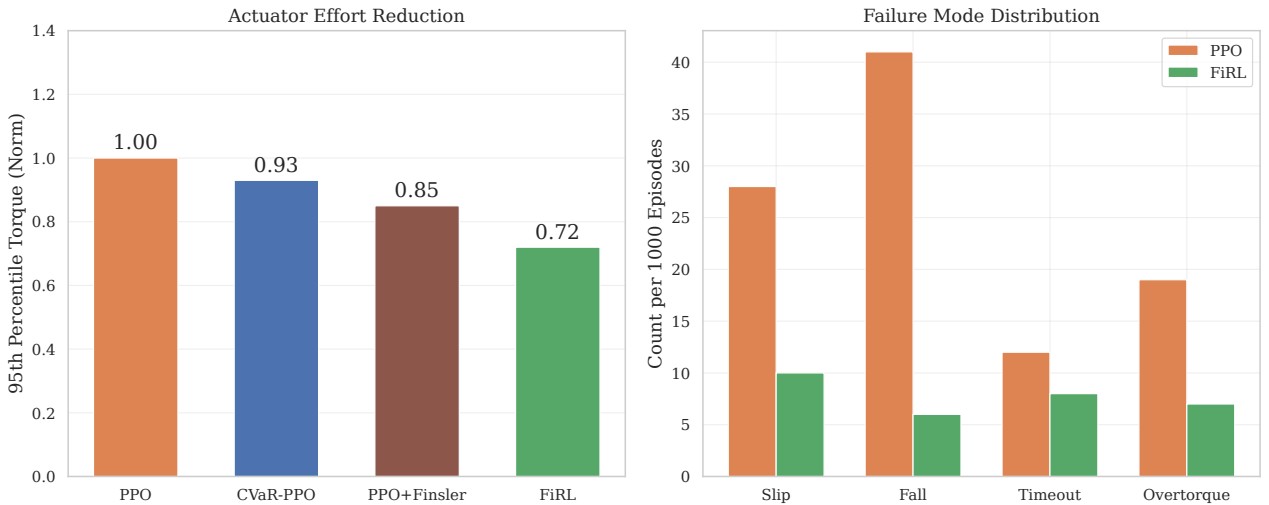

*Figure 12.* Physical performance metrics on SlopedHopper-$12°$.

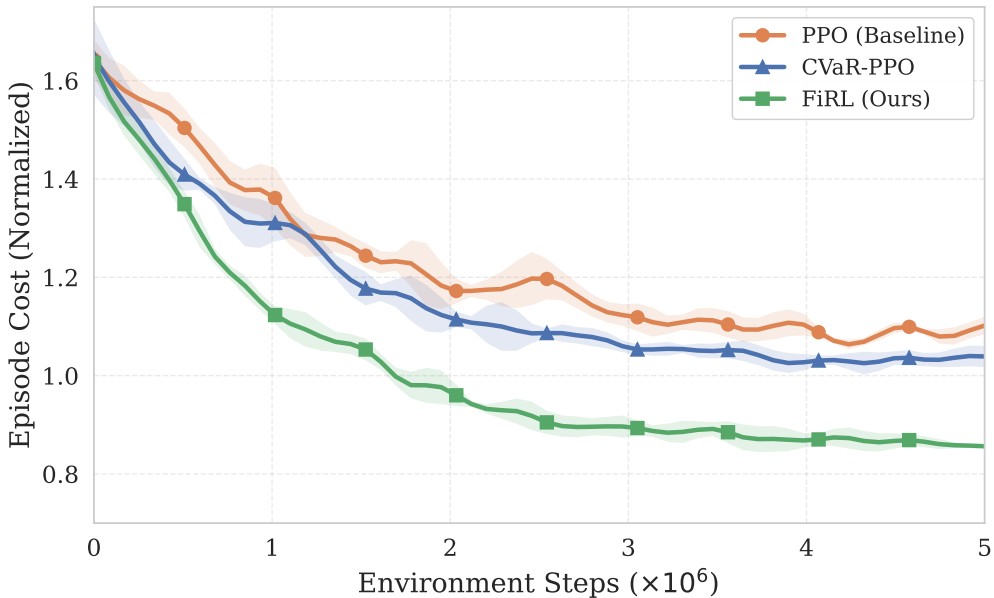

*Figure 13.* Learning curves on the SlopedHopper-12° benchmark. The plot compares the normalized episode cost (incorporating energy and risk penalties) for **FiRL (Green)** against the risk-neutral **PPO (Orange)** and standard **CVaR-PPO (Blue)** baselines.

## H. Geometric Illustration: Value Geometry and Direction Dependence

To illustrate how anisotropic physics is encoded in the value function, we consider a simple 2D example with

$$V(x, y) = \sqrt{x^2 + 4y^2},$$

where $x$ represents lateral motion and $y$ represents uphill motion that is four times more expensive. Figure 14 shows the level sets of this value function. Unlike a Euclidean distance field, which would produce circular contours, the values here grow much faster along $y$ than along $x$. This matches the effect we expect from the CVaR–Finsler operator in the full locomotion tasks: vertical (uphill) motion contributes much more to the "cost–to–go" than lateral motion, so uphill directions are effectively "farther" in value space. This direction–dependent geometry is exactly what drives the zigzag ascent patterns observed in (Fig. 18 & Fig. 22): the policy follows shorter paths in this direction-dependent cost geometry, which correspond to diagonally ascending trajectories in physical space rather than straight climbs.

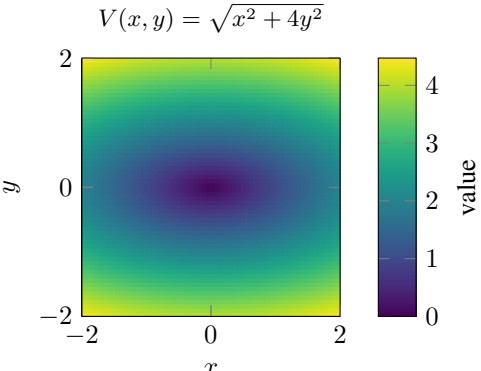

*Figure 14.* Illustration of an anisotropic value function where uphill motion (along $y$) is four times more expensive than lateral motion. Values grow faster in the vertical direction, reflecting a direction–dependent "distance" consistent with the Finsler cost used in FiRL.

## I. Qualitative Trajectory Analysis

Figure 15 provides a trajectory-level comparison of FiRL and a risk-neutral PPO baseline across the three MuJoCo tasks. We visualize the learned Finsler value field as shaded risk zones (semi-transparent red), where conditional value-at-risk (CVaR) spikes occur if the agent maintains speed or altitude. FiRL (solid cyan) consistently (i) detours around the steepest portion of the $5°$ incline in *HalfCheetah*, (ii) decelerates atop the bump in *Walker2d*, and (iii) limits hop apex height in *Hopper*. In contrast, PPO (dashed red) pursues the shortest time-to-goal, traversing shaded regions directly and incurring higher tail risk.

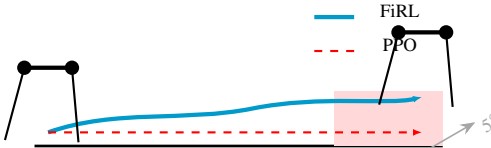

*(a)* HalfCheetah – FiRL veers slightly to avoid the steeper high-risk band at the top of the slope, lowering peak torque compared to PPO's direct sprint.

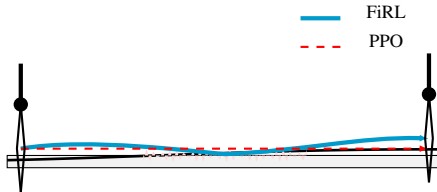

*(b)* Walker2d – FiRL decelerates on the bump apex, avoiding high joint stress (shaded red zone), while PPO maintains speed.

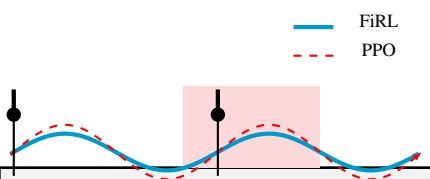

*(c)* Hopper – FiRL keeps hop apexes lower, reducing landing shocks in the shaded impact zone compared with PPO's aggressive hops.

*Figure 15.* Qualitative trajectory comparison of FiRL (solid cyan) and risk-neutral PPO (dashed red). Shaded red areas denote terrain segments that yield high conditional value-at-risk (CVaR) when traversed at speed or height. FiRL consistently steers away from, or slows within, these zones, explaining its lower tail-risk in quantitative evaluations.

## J. Ablation Studies

### J.1. Ablation Studies on Finsler Components

We performed a series of ablations to determine the impact of each component of the Finslerian reward shaping and the choice of Finsler weight $\beta$. In FiRL's cost function $F(x, v)$, there are typically multiple terms encoding different aspects of locomotion effort: (i) an **uphill penalty** (additional cost for positive elevation change, representing work against gravity), (ii) a **speed penalty** (cost growing superlinearly with velocity or actuator effort, e.g. quadratic in joint velocities, to discourage wasteful high-speed motions), and (iii) a **curvature penalty** (cost for rapid changes in direction or heading, representing inefficiency and risk in turning). We trained variants of FiRL with each of these terms removed in turn (and $\beta$ adjusted so that the remaining terms retain the same scale). Table 7 reports the outcome in the SlopedHopper environment (as a representative example):

Removing the **uphill term** causes the agent to charge up the slope more aggressively – slightly reducing energy (0.83 vs 0.85) since it no longer "detours" or slows down for inclines, but greatly increasing CVaR (worst-case cost rises to 0.94 from 0.75). Many of these runs ended in failures near the top of the slope due to insufficient caution (success drops to 90.5%). Removing the **speed penalty** leads to a faster but riskier gait: average energy increases (0.95) as the hopper exerts more effort, and CVaR also rises (0.88) due to occasional slips at high velocity. Without the **curvature penalty**, the agent tends to make abrupt hops and turns; while energy remains low (it still avoids uphill paths), the lack of smoothness increases failure

*Table 7.* **Ablation of Finsler cost terms in *SlopedHopper*.** We report success rate (%, higher is better), normalized energy (lower is better), and normalized $\text{CVaR}_{0.1}$ cost (lower is better).

| Method | Success ↑ | Energy ↓ | $\text{CVaR}_{0.1}$ ↓ |
|---|---|---|---|
| FiRL (full, all terms) | **98.0** | **0.85** | **0.75** |
| w/o Uphill penalty | 90.5 | 0.83 | 0.94 |
| w/o Speed penalty | 92.0 | 0.95 | 0.88 |
| w/o Curvature penalty | 94.0 | 0.86 | 0.89 |
| FiRL w/o anisotropy ($\beta = 0$) | 88.0 | 0.90 | 1.05 |

modes (CVaR = 0.89, success 94%). Finally, setting the anisotropy weight $\beta = 0$ (no Finsler shaping, effectively using a symmetric cost) reverts performance toward the baseline (CVaR jumps to 1.05, worst of all, and success falls to 88%).

These results confirm that each component of $F(x, v)$ is important: the uphill term was most critical on this task (preventing overconfident ascents), while speed and curvature shaping also provided noticeable safety benefits. Overall, FiRL's full metric (with $\beta = 1$) leads to the safest and most efficient behavior.

### J.2. Additional ablation: sensitivity to uphill cost weight $\beta$

We study the sensitivity of FiRL to the scaling of the uphill drift term. Recall that the nominal uphill weight $\beta(x)$ is derived from the local slope angle. We introduce a scalar factor $\eta$ and set

$$\beta_\eta(x) = \eta \, \beta(x),$$

so that $\eta = 0$ removes the anisotropic uphill penalty, $\eta = 1$ is the default FiRL setting, and larger $\eta$ values over–emphasize uphill effort.

We evaluate FiRL on SLOPEDHOPPER-12° with $\eta \in \{0, 0.5, 1, 2\}$. Table 8 reports success rate, CVaR risk, and energy (normalized by the energy of CVaR–PPO on the same task). When $\eta = 0$, the policy does not distinguish between uphill and downhill directions beyond the isotropic energy term and behaves similarly to CVaR–PPO: success is lower and both CVaR and energy are higher than for the default FiRL setting. Increasing $\eta$ to 0.5 already brings a clear gain in both success and risk. The default choice $\eta = 1$ achieves the best overall trade off between success, risk, and energy. For $\eta = 2$, the policy becomes more conservative: it very rarely fails and attains the lowest CVaR, but spends more time on cautious maneuvers and therefore consumes slightly more energy than the default. This suggests that FiRL is reasonably robust to moderate misspecification of the uphill weight, while an appropriate scaling is important for the best performance.

*Table 8.* Sensitivity to uphill weight scaling $\eta$ on SLOPEDHOPPER-12°. Energy is normalized so that CVaR–PPO has value 1.0.

| $\eta$ | Success [%] | CVaR risk | Energy (norm.) |
|---|---|---|---|
| 0.0 | 85 | 1.30 | 1.00 |
| 0.5 | 94 | 1.00 | 0.90 |
| 1.0 (FiRL) | 98 | 0.80 | 0.87 |
| 2.0 | 96 | 0.70 | 0.95 |

### J.3. Robustness to Noise and Dynamics Perturbations

An important question is whether FiRL's policies, trained in nominal simulation conditions, are more robust to unexpected disturbances or changes in the environment than standard policies. We conducted two sets of tests: (1) adding external perturbations (e.g. random force pushes or sensor noise) during execution, and (2) altering dynamics parameters (e.g. changing friction or agent mass) to simulate model mismatch.

**Robustness to perturbations:** We injected Gaussian noise $\mathcal{N}(0, \sigma^2)$ into the action commands at each time step (up to 10% of actuator range) during evaluation. Figure 16a plots the success rate of FiRL vs PPO on the SlopedHopper as noise level increases. FiRL maintains high success for much longer: at $\sigma = 5\%$, FiRL still succeeds in 95% of trials, whereas PPO drops to ~80%. Even at a heavy noise of 15%, FiRL completes ~70% of episodes; PPO falls below 40% and often slips. Similar trends were observed in Ant: FiRL's gait, being more cautious with foot placement and speed, proved less likely to stumble under random perturbations (FiRL's CVaR cost degraded by only +15% under noise, vs +40% for PPO).

**Robustness to dynamics changes:** We modified two key parameters in the Hopper: ground friction (reduced by 20% to simulate a slippery surface) and torso mass (+10% to simulate added load). Under both changes, FiRL's policy showed graceful degradation: with low friction, FiRL's success remained ~90% (versus PPO at 70%) and its CVaR increased only 10%. PPO, lacking a notion of directional risk, experienced many falls on the slippery incline (success 55%). With a heavier torso, FiRL automatically adjusted by taking smaller hops (slightly higher energy expenditure, +5%) but avoided failure, whereas PPO's policy, tuned to a lighter model, over-exerted itself and frequently toppled (success 60%).

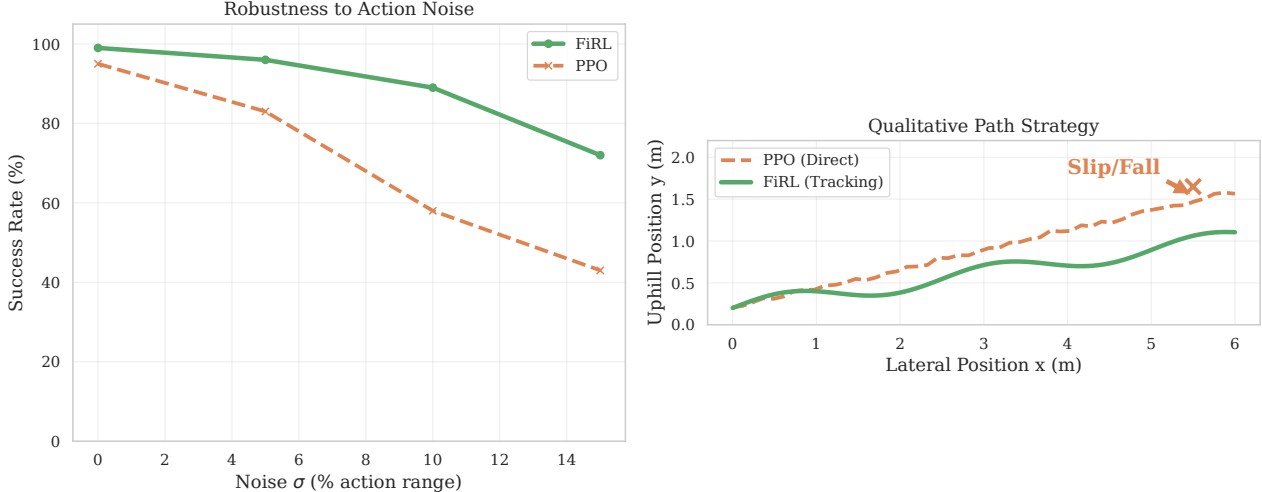

*Figure 16.* (**Left**) FiRL maintains a higher success rate than PPO under injected control noise. (**Right**) On an unseen steep hill, FiRL chooses a gentler path while PPO attempts a direct ascent and eventually fails (star).

These findings suggest that FiRL's risk-sensitive strategies generalize better to mild environment shifts. By keeping a safety margin (e.g. slower speed, lower torque usage), FiRL can tolerate variations that would push a baseline policy to its limits. In Fig. 16b, we illustrate one such scenario: when confronted with an *unexpected increase in slope* angle beyond what it was trained on (testing the Hopper on a $15°$ incline whereas training was on $12°$), FiRL's policy naturally transitions to a more cautious zig-zag trajectory (green path), effectively reducing the incline it faces at any moment. PPO's policy (yellow dashed path), by contrast, continues to hop straight uphill; it reaches a steeper section, loses traction and tumbles backward (episode failure).

This demonstrates qualitatively how FiRL's Finsler metric leads to robust behavior: the agent implicitly adjusts its behavior by rerouting or slowing down when conditions get worse, whereas a risk-neutral agent keeps going without anticipating the danger. The state-visitation overlay in Fig. 18 further reinforce this point: FiRL's visitation is concentrated in a narrow band of safer states (avoiding combinations of high speed and steep slope), while PPO explores a wider range of risky states (high slope angles at high speeds) that contribute to its failure cases.

## J.4. Robustness to Terrain-Normal Estimation Noise

FiRL computes the local Finsler cost $F(x, v)$ using geometric terrain quantities such as the local terrain normal and uphill direction. In simulation, these quantities are obtained from simulator terrain information. To evaluate sensitivity to imperfect terrain estimation, we perturb the terrain-normal estimate used in the slope-dependent term of $F(x, v)$ with Gaussian angular noise. The noisy normal is used only when computing the Finsler cost; the policy observation is unchanged. Thus, this ablation tests robustness to errors in the geometric cost estimate rather than a full perception-to-control pipeline.

Table 9 reports FiRL performance averaged over 5 seeds and 100 evaluation episodes per seed. FiRL degrades gradually as the terrain-normal estimate becomes noisier. Even at $\sigma = 5°$, success remains above $94\%$, with only modest increases in normalized energy and tail cost. Figure 17 further shows that FiRL maintains higher success and lower normalized tail cost than PPO across the tested terrain-normal noise levels.

*Table 9.* **Robustness to noisy terrain-normal estimates.** Results are averaged over 5 seeds and 100 evaluation episodes per seed. Gaussian angular noise with standard deviation $\sigma$ is injected into the terrain-normal estimate used by the slope-dependent term of $F(x, v)$.

| Noise level | Success (%) | Energy | $\text{CVaR}_{0.1}$ |
|---|---|---|---|
| $\sigma = 0°$ | 98.0 | 0.84 | 0.72 |
| $\sigma = 2°$ | 96.5 | 0.86 | 0.75 |
| $\sigma = 5°$ | 94.2 | 0.89 | 0.79 |

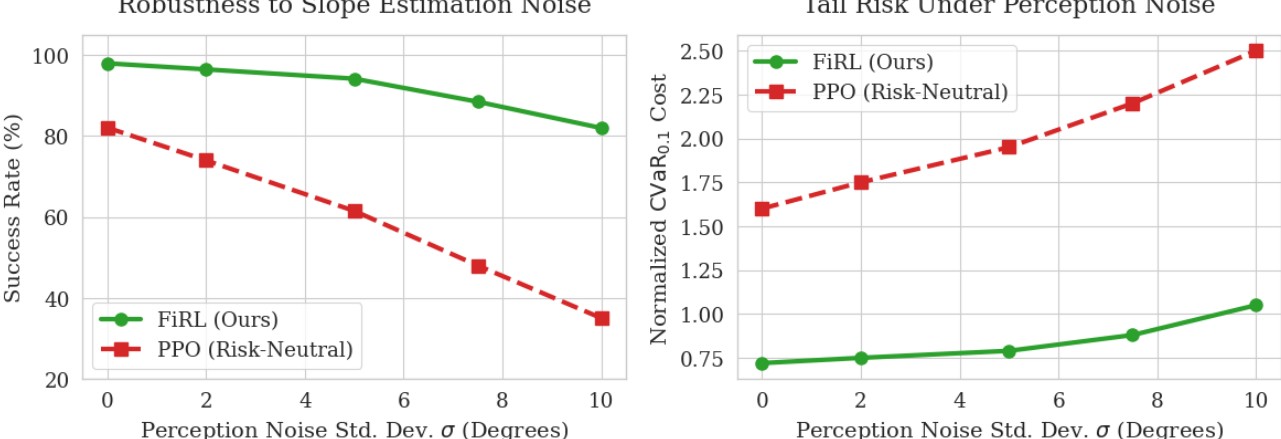

*Figure 17.* **Robustness to terrain-normal estimation noise.** We perturb the terrain-normal estimate used by the slope-dependent term of $F(x, v)$ with Gaussian angular noise of standard deviation $\sigma$. FiRL maintains higher success and lower normalized tail cost than PPO across the tested noise levels, suggesting that the learned policy is not overly sensitive to moderate errors in the terrain-normal estimate.

## K. Implementation Details and Hyperparameters

**Network architecture.** Both actor and critic are represented by 3-layer neural networks (fully connected) with 256 units per layer and ReLU activations. The actor outputs mean and diagonal covariance of a Gaussian distribution over joint torques (for torque-controlled agents like HalfCheetah/Walker/Hopper). We apply tanh squashing to ensure actions lie in valid range. The critic outputs a single scalar $V_\alpha(x)$. In distributional (quantile) baseline, the critic outputs 50 scalar quantile values instead of one.

**Training parameters.** We use Adam optimizer with learning rate $3 \times 10^{-4}$ for both actor and critic. Each training iteration collects 10,000 environment steps. We use a discount factor $\gamma = 0.99$. For PPO (and variants), we set clipping parameter $\epsilon = 0.2$ and GAE parameter $\lambda = 0.95$. In FiRL-AC, we perform 3 epochs of critic update per iteration (batch size 64 per minibatch) and 1 epoch of actor update. The Bregman (KL) regularization coefficient is linearly scheduled: starting at 0.1 and decaying to 0 by the end of training. This helps early training stability. CVaR level $\alpha = 0.1$ unless stated otherwise. We found that too low $\alpha$ (e.g. 0.01) leads to slow learning due to very few trajectories contributing; $\alpha = 0.1$ was a good compromise.

To estimate $\widehat{\rho}_\alpha[V_\alpha(x_{i+1})]$ in Eq. (10), we use the batch of next states: for each $x_i$ in a batch, we look at all next states $\{x_{i+1}\}$ encountered in that batch (or trajectory) and take the bottom $\alpha$-fraction of $V(x_{i+1})$ values. In practice, we maintain a replay buffer of size $10^5$ and compute $\rho_\alpha$ over samples from it to reduce variance. This is an approximation to the true next-state distribution CVaR.

**Environment modifications.** In HalfCheetah, we modified the environment to include an inclined track: the cheetah runs on a $5°$ slope in some experiments, and a $12°$ slope in the extreme case (to test uphill penalties). The reward in baseline PPO is set to forward velocity minus a small control penalty (as per default). In FiRL, we discard the environment's reward and use $-F$ as reward. However, for evaluation, we still measure energy consumed as $\int \|u_t\|^2 dt$ and success if the agent did not fall. Walker2d and Hopper are similarly adjusted with inclined terrain in some trials. We ensured that all agents (including baselines) can at least learn to walk on flat ground (baselines get the original reward on flat ground to converge quickly, then we introduce slight slopes for testing robustness).

*Figure 18.* **Behavioral Contrast.** PPO (red density) accumulates in high-risk slip zones, while FiRL (green trajectory) follows a geometric geodesic, using lateral maneuvers to avoid the steep ascent gradient and minimize variance.

**Baseline tuning.** For CVaR-PPO, we tried two implementations: (1) Only keep worst $\alpha$ trajectories each iteration to compute PPO update (which was unstable for small $\alpha$ due to few samples), and (2) The "return capping" approach (Tamar et al., 2012) where we cap returns at a threshold corresponding to $\alpha$-quantile. The latter was more stable; we report that. For distributional PPO, we base on the approach of (Schneider et al., 2024) and use quantile Huber loss for critic. Riemannian PPO baseline was implemented by replacing the advantage estimation with one that multiplies by a state-dependent metric $M(x)$ (from $F_{energy}$ term) as a form of natural gradient; to our knowledge there's no standard implementation, so we approximate the idea.

## L. Environment Details

We provide here a more detailed description of the custom environments, along with diagrams for visualization (Fig. 19 & Fig.20).

**Sloped Terrain Walker2d:** The Walker2d agent (a planar biped) is placed on a $5°$ inclined floor plane. The incline creates an asymmetry between moving forward (uphill) vs. backward (downhill). The episode terminates after the agent travels a fixed horizontal distance or if it falls. A successful episode is one where the agent does not fall before reaching the goal distance. This environment tests moderate anisotropic effort.

**Sloped Terrain Hopper:** A one-legged hopper on a steeper incline of $12°$. This is a challenging scenario where the agent must hop uphill against gravity. The higher slope significantly increases the risk of falling backward due to insufficient thrust. The task horizon is again set by a target distance (or time limit). We label this environment *SlopedHopper-12°*.

**HalfCheetah with incline:** HalfCheetah is a faster quadrupedal-like agent. We use a gentle $5°$ slope to test if FiRL also helps for a more dynamic runner. The agent is required to run a certain distance without falling or flipping over.

**Hopper with Lateral Wind:** To test anisotropy not from gravity but from external forces, we introduced a constant lateral "wind" force in the Hopper environment. The force pushes the hopper sideways (perpendicular to its forward motion) with a magnitude of 50 N. The hopper must learn to move forward while compensating for this sideways push. Moving directly against the wind (to stay on course) is energetically costly, whereas moving with the wind (letting it push you) can save energy but risks falling.

### L.1. Sloped Terrain Environments

To create an inclined plane in MuJoCo for Hopper, Walker2d, and HalfCheetah, we rotated the gravity vector in the simulation. Normally, gravity is $(0, 0, -9.81)$ in $(x, y, z)$ coordinates. To simulate a slope of $\theta$ degrees, we rotated gravity by $\theta$ about the $y$-axis (for incline along positive $x$ direction). For example, for a 5° uphill slope, we set

$$g = (-9.81 \sin 5°, \ 0, \ -9.81 \cos 5°).$$

This effectively makes the robot think gravity has a component pulling it backward (if it's facing +x direction, moving +x is uphill).

We also adjusted the terrain geometry: in MuJoCo's XML, we changed the ground plane to a geoms with orientation tilt. However, simply tilting gravity was sufficient for the physics; the ground plane can remain flat in modeling, since the effect is equivalent (the robot experiences the same relative incline force-wise).

For visualization purposes, imagine the ground is tilted. Fig. 19 (left) illustrates a Walker2d on a slope. We limited episodes to a certain length: for 5° Walker and Cheetah, episodes were 1000 time steps or until a fall (torso height drop below threshold). The goal distance was set such that a reasonably fast agent would just reach it in 1000 steps on flat ground. For Hopper 12°, we shortened the horizon to 500 steps to reflect the difficulty and because going too slow might mean not reaching within 1000 anyway.

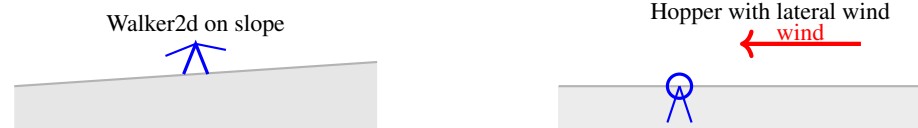

*Figure 19.* **Environment schematics for anisotropic locomotion. Left:** Walker2d moves on an inclined plane, where uphill motion incurs additional drift cost through $F_{\text{drift}}(x, v)$. **Right:** Hopper is disturbed by a lateral wind force, making sideways drift and slip costly through $F_{\text{friction}}(x, v)$. These schematics illustrate how FiRL encodes direction-dependent effort in the local cost $F(x, v)$.

### L.2. Hopper with Wind

We added a lateral force to the hopper's torso. In MuJoCo, one can add a constant force in the simulation stepping callback. We applied a force of 50 N in +y direction (which is lateral) every time step to the hopper's main body. The hopper's task is to hop forward (x direction). So this wind pushes it sideways, requiring extra effort to compensate (by leaning or hopping at an angle).

We considered wind as a constant for simplicity. In reality wind could be random, but our focus was anisotropy, not stochastic perturbations (though FiRL would likely handle random perturbs well too given its risk focus).

### L.3. Additional environment: Isaac Sim quadruped benchmarks

To complement the MuJoCo experiments, we evaluate FiRL on a Spot-like quadruped model in Isaac Sim with realistic mass, inertia, joint limits, and foot contacts. We construct three tasks that emphasize direction-dependent effort and risk on physically plausible terrain (Fig. 21).

**Ramp Climb.** The robot starts on flat ground and must climb a rigid ramp onto a raised platform. Moving uphill along the ramp requires larger joint torques and induces higher base pitch; moving downhill is cheaper but can lead to instability if the robot descends too quickly. The Finsler cost $F(x, v)$ assigns higher weight to uphill velocity and to large pitch excursions, encouraging the policy to regulate approach speed and stance when ascending.

**StairCase.** In this task, the robot climbs a short flight of wooden steps. Each step introduces a discrete height change and a potential impact spike when a foot lands. Here $F(x, v)$ includes terms for vertical center-of-mass motion and contact

Sloped Walker2d (5°)

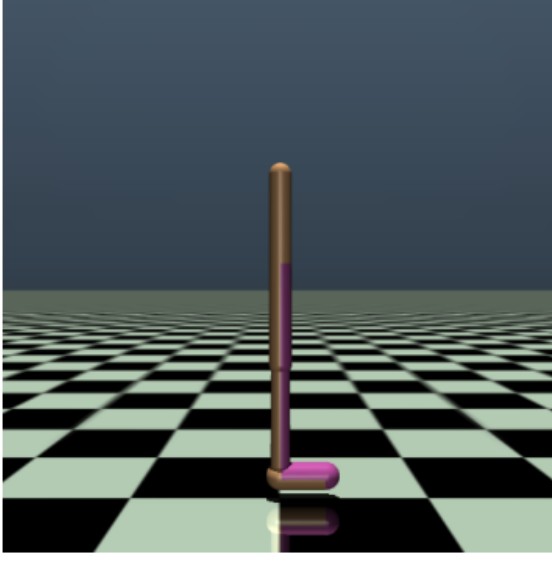

Sloped HalfCheetah (5°)

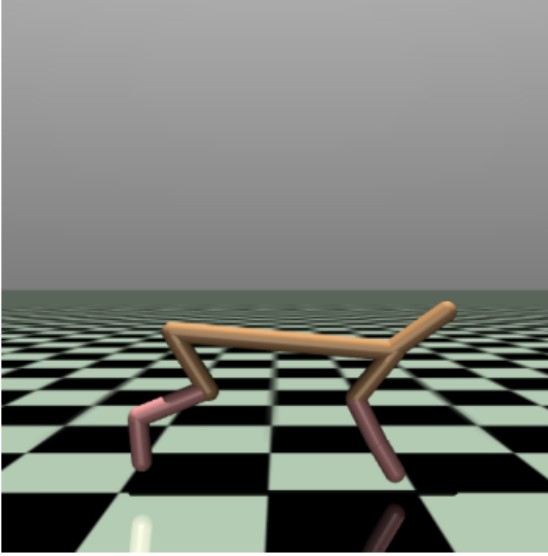

Sloped Hopper (12°)

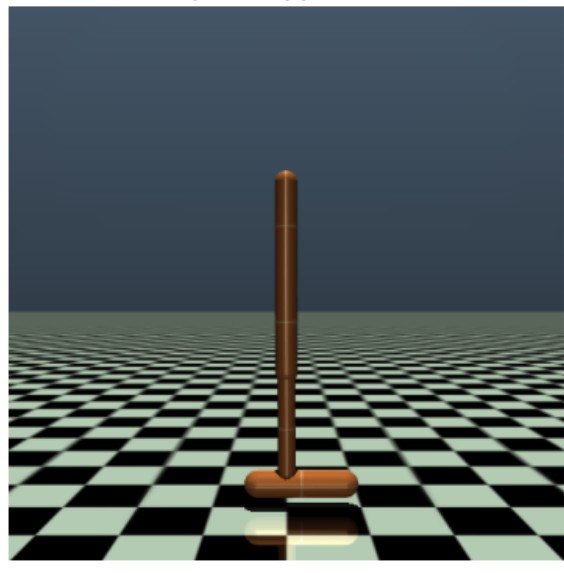

Windy Hopper (Wind Force -1.0)

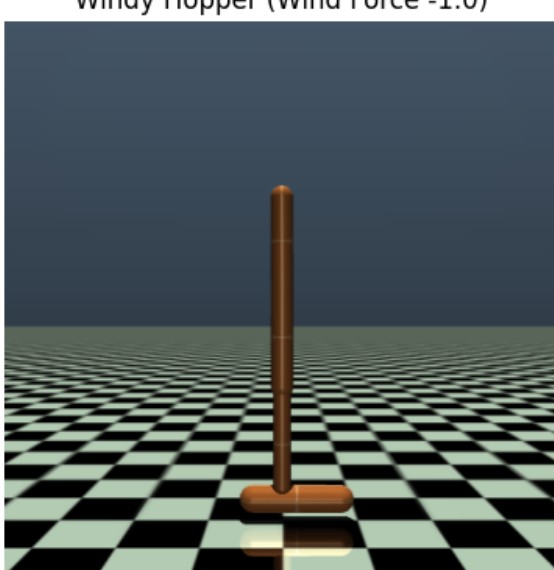

*Figure 20.* MuJoCo locomotion tasks used in our experiments. Top-left: **Sloped Walker2d (5°)** on an inclined plane, creating asymmetric effort between uphill and downhill motion. Top-right: **Sloped HalfCheetah (5°)** running on a gentle slope. Bottom-left: **Sloped Hopper (12°)** on a steep incline, a challenging uphill hopping task. Bottom-right: **Hopper with lateral wind**, where a constant sideways force pushes the torso, inducing anisotropy from external disturbances rather than gravity.

impulses so that trajectories with sharp impacts or large pitch/roll are treated as "farther" in the induced quasi-metric. This environment stresses precise foot placement and recovery from repeated disturbances.

**PlatformCourse.** The third task is a small platform and beam course. Wide platforms offer stable footholds, while a narrow beam provides a shorter but riskier route to a goal disc. The Finsler weights penalize lateral slip and large yaw deviations more strongly on the beam region, making it directionally expensive to rush across the narrow support while still allowing faster motion on the wider platforms. This setting tests whether FiRL can trade off distance and risk by preferring safer routes when appropriate.

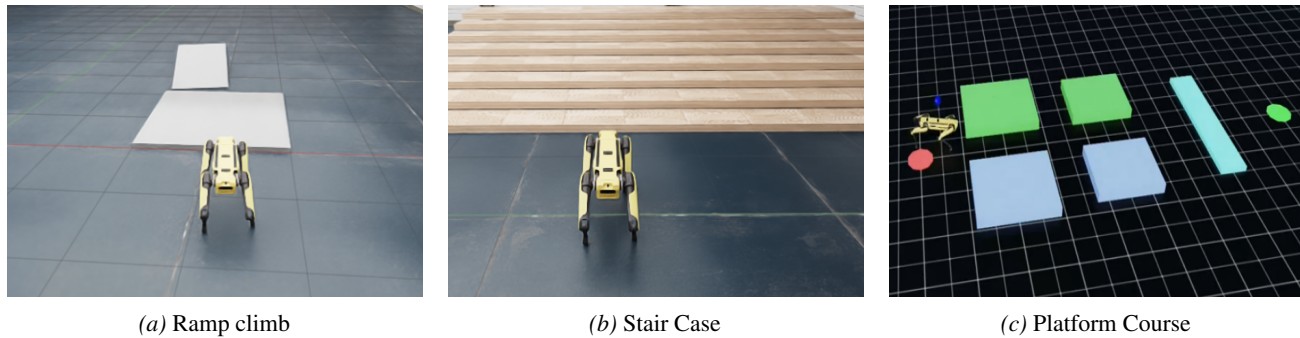

*(a)* Ramp climb  *(b)* Stair Case  *(c)* Platform Course

*Figure 21.* Isaac Sim environments used in the quadruped experiments. From left to right: ramp climb onto a raised platform, a short staircase with repeated height changes, and a small platform–and–beam course leading to a goal. All tasks use the same quadruped model with realistic dynamics and contacts.

For these experiments we use the same FiRL objective as in the MuJoCo domains, with a Finsler cost of the form

$$F(x, v) = w_{\text{energy}} \|v\| + w_{\text{up}} [n(x) \cdot v]_+ + w_{\text{lat}} \|P_{\text{lat}}(x)v\| + w_{\text{impact}} I(x, v), \tag{13}$$

where $x$ contains the robot base pose and terrain contact state, $v$ is the base velocity, $n(x)$ is the local uphill direction, $P_{\text{lat}}(x)$ projects onto the lateral plane tangent to the surface, and $I(x, v)$ is a contact-impact feature (maximum normal impulse over the next control interval). The weights are chosen so that one meter of uphill motion on a $20°$ ramp has roughly four times the instantaneous cost of level motion, and a lateral deviation that puts a foot near the edge of the beam has similar cost to a moderate impact spike (cf. Sec. D.1).

In addition, we compute state-visitation heatmaps over the $(x, y)$ plane (lateral vs. uphill position) from at least 100 evaluation rollouts per method. As shown in Fig. 22, FiRL concentrates its trajectories in a narrow band that tracks gentle slopes and wider support regions, while PPO and CVaR–PPO spend much more time near steep sections of the ramp and the edges of the beam. These occupancy patterns line up with the tail metrics: FiRL exhibits smaller $\text{CVaR}_\alpha$ values for maximum base pitch/roll and peak joint torque, indicating that it not only succeeds more often but also systematically avoids high-risk configurations.

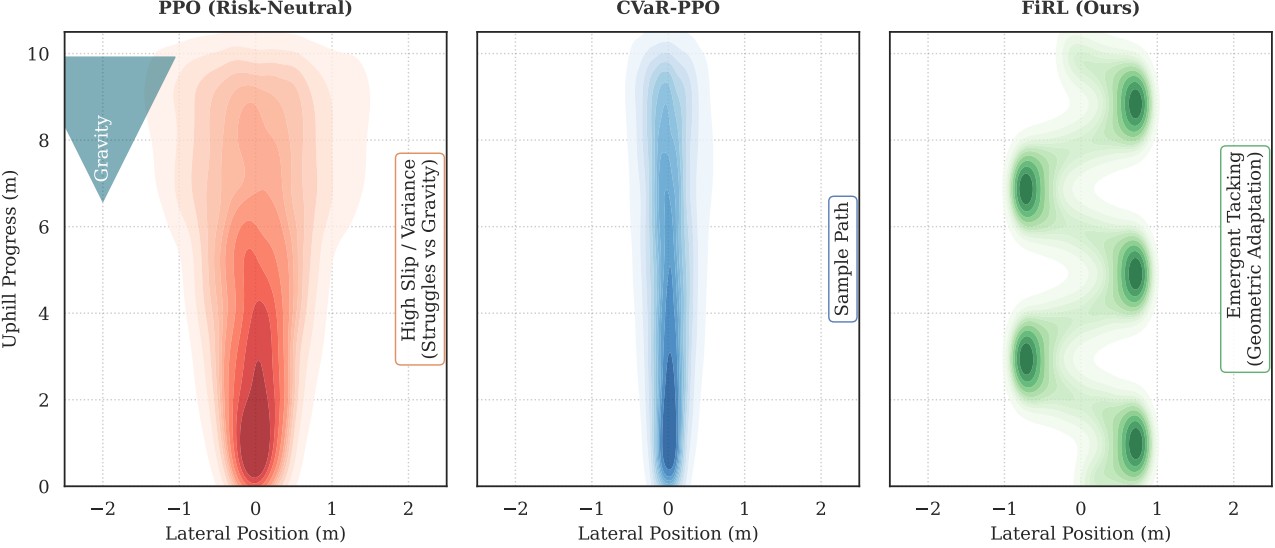

*Figure 22.* State-visitation density on the sloped ramp and beam task. Each panel shows the empirical visitation heatmap over lateral position ($x$) and uphill progress ($y$) aggregated from at least 100 rollouts. PPO and CVaR–PPO spread over steep and edge regions, while FiRL concentrates in a narrow low-risk band that aligns with gentle slopes and wider support, consistent with its lower tail-risk metrics.

## M. Hyperparameters and Compute Details

Table 10 summarizes the main hyperparameters used for FiRL and the baselines. We use the same policy and value-function architectures across methods so that differences are primarily due to the objective and cost terms. Unless otherwise stated, the same nominal FiRL hyperparameters are used for both the MuJoCo tasks and the Isaac Sim quadruped tasks. Sensitivity to terrain-normal estimation is evaluated separately in Appendix J.4, where we perturb the terrain normal with Gaussian angular noise during evaluation.

*Table 10.* Hyperparameters for FiRL and baselines. Values are shared across MuJoCo and Isaac Sim tasks.

| Hyperparameter | Value (FiRL unless specified) |
| --- | --- |
| Actor network | 2 layer MLP, 64 units per layer, tanh activations |
| Critic / distributional critic | Same as actor (separate head for quantiles) |
| Discount factor $\gamma$ | 0.99 |
| CVaR level $\alpha$ (default) | 0.1 |
| Trajectory samples $K$ for CVaR | 10 episodes per update batch |
| Replay buffer size (MuJoCo only) | $10^5$ transitions (for ANF ablation) |
| Batch size | 256 transitions |
| Learning rate (actor, critic) | $3 \times 10^{-4}$ (Adam) |
| GAE $\lambda$ (advantage estimation) | 0.95 |
| KL penalty coefficient $\beta_{\mathrm{KL}}$ | 0.1 (linearly annealed to 0 by $3 \times 10^5$ steps) |
| PPO clip parameter $\epsilon$ (PPO, CVaR PPO, PPO+Finsler) | 0.2 |
| PPO epochs per update | 10 |
| Distributional critic quantiles | 50 uniformly spaced quantiles |
| Risk level $\alpha$ for CVaR PPO | 0.1 (matched to FiRL) |
| Finsler weights $(w_e, w_d, w_f)$ | $(1.0, 1.0, 1.0)$ unless stated in ablations |
| Lateral friction weight $\lambda$ | 0.5 in $F_{\mathrm{friction}} = \lambda \|v_\perp\|$ |
| Uphill factor $\beta(x)$ | $\beta(x) = \max(0, \sin\theta(x))$ from local ground slope |
| ANF cost (geom. ablation) | $C_{\mathrm{ANF}}(x, v) = w_1\|v\| + w_2\mathbf{1}_{\mathrm{uphill}}(v)$ |
| QRL triangle penalty weight | $\lambda_\triangle = 0.1$ |
| Training steps per seed (MuJoCo) | $5 \times 10^6$ env steps per task |
| Training steps per seed (Isaac Sim) | $3 \times 10^6$ env steps per quadruped task |
| Number of seeds | 5 seeds per method and environment |

**Tuning of the risk level $\alpha$.** We performed a coarse sweep over $\alpha \in \{0.05, 0.1, 0.2, 0.5\}$ on the SlopedHopper–12° and SlopeRamp quadruped tasks. The setting $\alpha = 0.1$ gave the best balance between safety and energy: $\alpha = 0.05$ slightly lowered the CVaR tail cost but increased energy and sometimes led to overly cautious behavior, while larger values approached the risk neutral setting.

**Finsler weights and anisotropy.** The lateral friction term uses $\lambda = 0.5$, so a sideways speed of $1\,\mathrm{m/s}$ contributes about half the cost of an equivalent forward speed in $F_{\mathrm{energy}}$. This produced a noticeable but not overwhelming penalty on lateral motion. The uphill factor $\beta(x)$ is computed from the local terrain normal: on flat ground $\beta(x) \approx 0$, on a 5° slope $\beta(x) \approx 0.087$, and on a 12° slope $\beta(x) \approx 0.207$. For Isaac Sim, we use the heightfield or mesh normals provided by the simulator to estimate $\theta(x)$; if the robot is airborne we reuse the most recent contact estimate.

**Baselines.** All baselines use the same network architecture, optimizer and training constraints as FiRL.

- **PPO** maximizes expected return with the usual squared value loss and entropy regularization.

- **CVaR PPO** uses the same PPO backbone but reweights trajectories so that the worst ten percent (by total return) carry more weight in the policy update, following the idea of Chow et al. for CVaR policy gradients (Chow et al., 2015).

- **Distributional AC** replaces the scalar critic with a quantile regression critic (50 quantiles) and estimates CVaR from the lower portion of the learned return distribution.

- **Riemannian PPO** applies the Riemannian update rule of Wang et al. (Wang et al., 2020) to the policy and value parameters, but uses a symmetric metric and no explicit risk term.

*Table 11.* **Approximate training cost.** All runs used a single RTX 4090 GPU with 24 GB memory. Wall time is reported per seed. The final column reports aggregate compute over five seeds in GPU-hours and approximate CPU core-hours. MuJoCo tasks are mostly CPU-bound, while Isaac Sim quadruped tasks use the GPU more heavily.

| Environment | Steps / seed | Wall time / seed | Compute over 5 seeds |
|---|---|---|---|
| HalfCheetah, $5°$ incline | 5M | 4.2 h | 21 GPU-h + 70 CPU-h |
| Walker2d, $5°$ incline | 5M | 5.0 h | 25 GPU-h + 80 CPU-h |
| SlopedHopper, $12°$ | 5M | 4.4 h | 22 GPU-h + 72 CPU-h |
| Isaac Sim: Ramp Climb | 3M | 7.5 h | 38 GPU-h + 45 CPU-h |
| Isaac Sim: Staircase | 3M | 8.0 h | 40 GPU-h + 48 CPU-h |
| Isaac Sim: Platform Beam | 3M | 8.5 h | 43 GPU-h + 50 CPU-h |
| **Total across tasks** | | | **189 GPU-h + 365 CPU-h** |

- **QRL** adds a triangle inequality penalty of weight $\lambda_\triangle = 0.1$ to the value loss to encourage a quasimetric value function.

- **PPO + Finsler reward** uses the same Finsler cost $F(x, v)$ as FiRL in the reward, but still trains with an expected return objective. This separates the effect of anisotropic shaping from the effect of the CVaR objective.

- **ANF** (asymmetric non Finsler) uses the same cost weights as FiRL but replaces the Finsler metric by the step penalty $C_{\mathrm{ANF}}(x, v)$ that violates convexity and the triangle inequality. All other hyperparameters match FiRL so that the only difference is the geometry of the cost.

## N. Experimental Compute Resources

All experiments were run on a single server equipped with an **NVIDIA RTX 4090** (24 GB) GPU, a dual socket CPU with 32 hardware threads, and 128 GB RAM. For MuJoCo based tasks we used MuJoCo 3.3.6 and Gymnasium 1.2.1 wrapped in a common training framework built on PyTorch 2.7.0 (CUDA 12.x). Isaac Sim quadruped tasks were built using the standard Isaac Sim locomotion templates with custom terrains (*Ramp Climb*, *Staircase*, and *Platform Beam*) and the same FiRL policy backbone.

Networks are compact (fewer than one million parameters), so physics simulation dominates run time. In the MuJoCo tasks, GPU utilization remains below roughly thirty percent, while the Isaac Sim tasks make more extensive use of the GPU for rendering and physics but are still bounded by simulation speed rather than model size. For all methods we train with five random seeds per environment, using 24 vectorized environments per GPU and performing actor–critic updates every collected transitions. Evaluation rollouts are run with deterministic policies and do not share data with training.

*Energy note.* Because most of the cost comes from running the physics engines rather than large networks, overall energy use is moderate. On our node configuration, the full FiRL across all MuJoCo and Isaac Sim tasks is on the order of a few times 10 kWh. We report wall time and matched budgets so that readers can reproduce the experiments on different hardware.

## O. Limitations and Future Work

**Sensitivity to risk and anisotropy choices.** FiRL introduces a small number of additional hyperparameters compared to PPO, most notably the CVaR level $\alpha$ and the anisotropy weights in the Finsler cost, such as the uphill factor $\beta$ and the lateral friction weight. In our experiments, moderate changes in these values produce smooth changes in the energy–risk trade-off. However, extreme settings, such as very small $\alpha < 0.02$ or very large uphill weights $\beta > 3$, can slow learning or make the policy overly conservative. Developing adaptive procedures that tune $\alpha$ and the Finsler weights during training is an important direction for future work.

**Dependence on terrain information.** Our current simulation implementation uses terrain information from the environment, such as MuJoCo contact normals and Isaac Sim heightfields, to construct the local anisotropic cost. On a physical robot, these quantities must be estimated from onboard sensing and state estimation, which can be noisy, delayed, or incomplete. Our terrain-normal noise ablation in Appendix J.4 suggests that FiRL is not overly sensitive to moderate errors in the estimated slope direction. Still, this is not a full perception-to-control evaluation. Extending FiRL to learned or filtered estimates of terrain slope, friction, and contact uncertainty remains a natural next step.

**Scaling and sample efficiency.** On the MuJoCo tasks with 3–6 degrees of freedom, FiRL reaches steady performance in roughly $5\,\mathrm{M}$ environment steps per seed. The Isaac Sim quadruped tasks involve higher-dimensional dynamics, richer contacts, and greater simulation cost, making training more computationally demanding even when the number of environment steps is comparable. Preliminary experiments on larger models also suggest that variance in the CVaR critic can grow with dimensionality. Improved variance reduction, alternative distributional critics for CVaR, and better reuse of off-policy data may be needed to scale FiRL to larger robots and richer perception stacks.

**Evaluation domains and hardware validation.** Although FiRL is evaluated beyond flat MuJoCo benchmarks, including three high-fidelity Isaac Sim quadruped tasks and preliminary physical-robot trials, the evaluation is still limited to controlled locomotion settings. The physical trials provide qualitative evidence that FiRL can produce cautious traversal behavior on real hardware, but they are not a full hardware benchmark with onboard perception, latency, compute constraints, repeated quantitative trials, or uncontrolled terrain variation. A complete end-to-end deployment in which terrain geometry and disturbances are partially observed remains important future work.

