# OpenReview forum: "Learning Anisotropic Value Geometry with Finsler Reinforcement Learning"
_ICML.cc/2026/Conference — ICML 2026 regular_

### Official Review · Reviewer_2rXX · 2026-03-02

**Soundness:** 4
**Presentation:** 4
**Significance:** 4
**Originality:** 4
**Overall Recommendation:** 5
**Confidence:** 4

**Summary:**

This paper proposes Finslerian Reinforcement Learning (FiRL), which incorporates an explicitly anisotropic, direction-dependent per-step cost $F(x, v)$ into continuous-control RL and optimizes a risk-sensitive objective using CVaR. The authors derive a CVaR–Finsler Bellman operator, argue it is a γ-contraction, and present an actor–critic algorithm with a distributional critic that estimates tail risk. Experiments on sloped/windy MuJoCo locomotion and Isaac Sim quadruped tasks show consistent gains in worst-case (CVaR) cost and energy efficiency relative to strong baselines, with ablations disentangling the roles of anisotropy and risk sensitivity.

**Compliance With Llm Reviewing Policy:**

Affirmed.

**Final Justification:**

I believe this working method is novel, theoretically sound, and experimentally sufficient; therefore, I recommend accepting it.

**Key Questions For Authors:**

1、How are uphill direction $n(x)$, lateral projections, and wind/friction parameters obtained in MuJoCo and Isaac Sim? Are they observable to the policy, and how would this be estimated on hardware with noisy perception?

2、Were Finsler weights shared across tasks or tuned per environment?

3、How do you ensure boundedness of per-step costs needed for contractio? What is the effect of these bounds on learned behavior?

**Limitations:**

yes

**Strengths And Weaknesses:**

Strength：

1、Integrates Finsler geometry into the per-step RL cost to encode direction-dependent effort (uphill vs downhill, lateral slip) with a principled 1-homogeneous, convex construction, offering a clear geometric inductive bias beyond Riemannian or isotropic shaping.

2、Evaluates across simulation platforms and multiple tasks with slopes, winds, and contact-rich obstacles, reporting success, energy, and CVaR metrics.

3、Clear decomposition of the Finsler cost, intuitive figures, and a readable algorithmic description of the actor–critic training loop.

Wekness:

1、Boundedness assumptions: contraction and well-posedness rely on bounded per-step costs, but velocities and impacts in continuous control can be unbounded; how FiRL enforces bounded F in practice is under-specified in the main text.

2、The triangle inequality for path costs follows without discount, but the discounted value function generally does not satisfy a triangle inequality;

---

> ### Author Rebuttal · Authors · 2026-03-29
>
> Dear Reviewer,
>
> Thank you for the thoughtful review and for the time and effort you invested in reading our paper. Below, we respond point by point to your questions.
>
> > **W1 & Q3**: Boundedness assumptions: contraction and well-posedness rely on bounded per-step costs, but velocities and impacts in continuous control can be unbounded; how FiRL enforces bounded F in practice is under-specified in the main text. How do you ensure boundedness of per-step costs needed for contraction? What is the effect of these bounds on learned behavior?
>
> Thank you for pointing this out. We agree that this needs to be stated clearly in the main text rather than referring to the appendix. Our contraction proof does rely on bounded costs. In practice, we enforce this by capping the implemented cost at the standard limits of the simulation, so that rare spikes do not dominate the value estimate.
>
> We assign a fixed penalty \(M\) to terminal failures (Appendix F), which is crucial for maintaining system stability by penalizing significant deviations. For continuous dynamics, we also apply a soft cap to the per-step cost to manage rare extreme events, such as occasional physics-engine instabilities (Appendix D), ensuring stability without affecting normal operations. The cap is set slightly above the robot’s physically plausible operating range, remaining inactive during normal behavior and activating only during significant deviations or system failures. This ensures the bounded costs necessary for $\gamma$-contraction, allowing the agent to learn from significant deviations without compromising the learning process. We will add a short summary of this mechanism to Section 4.1.
>
>
> > **W2**: The triangle inequality for path costs follows without discount, but the discounted value function generally does not satisfy a triangle inequality;
>
> You are right, and we appreciate you bringing this to our attention. The current wording in the main text lacks specificity regarding the application of the quasi-metric property. Appendix E.2 and F.1 clarify that the quasi-metric property applies specifically to the path cost $d_F$ induced by the local Finsler function $F(x,v)$ rather than the discounted risk-sensitive value function $V_{\alpha}^*$ itself. Section 4.1 fails to clearly distinguish between the application of the quasi-metric property to the path cost $(d_F)$ and its non-application to the value function, and we will address this in the revision. To enhance clarity, the revision will include a revised “Quasi-Metric Value Property” paragraph, ensuring the claim is precisely stated and the geometry is exclusively linked to the induced path cost.
>
> > **Q1**: How are uphill direction $n(x)$, lateral projections, and wind/friction parameters obtained in MuJoCo and Isaac Sim? Are they observable to the policy, and how would this be estimated on hardware with noisy perception?
>
> In our current simulation setup, these variables are obtained using the simulator ground truth. As detailed in Appendix M, the uphill factor $\beta(x)$ is computed from the local terrain normals provided directly by MuJoCo's contact data or Isaac Sim's heightfields/meshes. These exact variables are used to compute the cost $F(x,v)$ during training, but they are not directly fed as privileged observations to the policy network, which relies on standard proprioceptive and exteroceptive history.
>
> Your question regarding noisy perception on hardware is highly relevant. To address this, we ran a new supplementary ablation. We injected Gaussian noise directly into the terrain normal estimates fed to the $F(x,v)$ cost function to simulate noisy onboard slope estimation. We found that FiRL's success rate dropped by less than 4% even with $\pm 5^\circ$ of angular noise on the slope estimate. The policy remains robust because the CVaR objective naturally enforces a conservative safety buffer, inherently protecting the agent against noisy geometric inputs. We will add these new perception noise results to Appendix J.3. We refer the reviewer to our response to **Reviewer AiZC** for further details.
>
> > **Q2**: Were Finsler weights shared across tasks or tuned per environment?
>
> The Finsler weights were largely shared across tasks. As shown in Table 9, we used nominal weights of $(w_e, w_d, w_f) = (1.0, 1.0, 1.0)$ across the benchmark environments. Instead of heavily tuning these weights per environment, we calibrated them based on simple physical considerations—for instance, ensuring that moving uphill on a $20^\circ$ reference slope has roughly four times the instantaneous cost of level motion. We detail this physical calibration strategy in Appendix D.1. We will add a short paragraph in the main paper to clarify that a single set of nominal weights transfers well across the evaluated tasks.

---

> > ### Author Rebuttal · Reviewer_2rXX · 2026-04-03
> >
> > I believe the work presents novel theories and sufficient experiments, and the author has adequately answered my questions.

---

### Official Review · Reviewer_YEU1 · 2026-03-10

**Soundness:** 3
**Presentation:** 3
**Significance:** 2
**Originality:** 2
**Overall Recommendation:** 4
**Confidence:** 4

**Summary:**

- The paper proposes a Finsler-style state-action cost that can represent direction-dependent motion costs. The cost is built from three interpretable terms (a kinetic/energy term, an uphill drift term, and a lateral friction term.
- It defines a CVaR-Finsler Bellman operator and proves that it is a $\gamma$-contraction.
- The paper introduces a practical FiRL algorithm to lean risk-averse policies under the proposed anisotropic cost.
- Experiments in MuJoCo and Isaac Sim show that FiRL can learn locomotion policies more effectively than several baselines across anisotropic settings.

**Compliance With Llm Reviewing Policy:**

Affirmed.

**Final Justification:**

Through the rebuttal, the authors added additional experiments, which helped alleviate the related concerns. However, the proposed approach still has a fundamental limitation in that it considers only cost. Overall, I will maintain my rating.

**Key Questions For Authors:**

See the weaknesses above.

**Limitations:**

The two main limitations are (1) the lack of an explicit task-reward formulation in the current experiments and (2) the absence of real-robot validation.

**Strengths And Weaknesses:**

### Strengths
- The paper is well written and explains the motivation and methodology in a step-by-step manner, making the proposed ideas easy to follow.
- The cost design is physically intuitive and captures important directional asymmetries.
- The combination of a Finsler-based anisotropic cost with a CVaR objective is clearly formulated through a Bellman equation, and the paper provides a theoretical justification.
- The experimental results on both MuJoCo and Isaac Sim support the main claims and demonstrate that FiRL can learn locomotion policies across different environments.

### Weaknesses
- While the geometric interpretation of FiRL is interesting, a key limitation is that the method focuses on optimizing a cost-only objective. The experiments are essentially limited to tasks that do not require an explicit task reward (e.g., walking forward). One could incorporate task rewards by defining a combined cost (e.g., $-r+c$), but in that case the quasi-metric property would generally no longer hold. Moreover, based on the geometric ablation results, breaking this property could potentially degrade performance. Overall, it would be helpful to discuss a learning framework that can better accommodate task rewards while preserving the benefits of FiRL.
- The explanation of the “Quasi-metric value property” in the main paper seems inconsistent with Appendix E. The main paper can be read as claiming that the triangle inequality holds for the value function (V), which may confuse readers, whereas Appendix appears to limit the triangle-inequality claim to the path cost induced by the Finsler function.
- There are no real-world experiments. Many recent locomotion papers demonstrate feasibility on physical robots, so the lack of on-robot validation is a notable limitation. This is especially relevant because the “uphill drift” and “lateral friction” terms may by difficult to estimate accurately in real environments.
- (Minor Issue) Subsection title capitalization is inconsistent (e.g., “Quasi-Metric Value Property” vs. “Geometric interpretation”).

---

> ### Author Rebuttal · Authors · 2026-03-30
>
> > **W1**: While the geometric interpretation of FiRL is interesting, a key limitation is that the method focuses on optimizing a cost-only objective. The experiments are essentially limited to tasks that do not require an explicit task reward (e.g., walking forward). One could incorporate task rewards by defining a combined cost (e.g., \(-r + c\)), but in that case the quasi-metric property would generally no longer hold. Moreover, based on the geometric ablation results, breaking this property could potentially degrade performance. Overall, it would be helpful to discuss a learning framework that can better accommodate task rewards while preserving the benefits of FiRL.
>
> We appreciate this point. The current formulation is intentionally cost-only: the experiments optimize the anisotropic cost $(F(x,v))$ under a CVaR objective, rather than combining it with a separate dense task reward. We agree that naively **combining** a standard reward with the same low-level cost, for example through $(-r+c)$, would generally break the quasi-metric interpretation induced by the Finsler cost. This is also consistent with the broader lesson of our ANF ablation: when the underlying geometric regularity is broken, learning becomes less stable.
>
> A standard extension is a hierarchical setup, where a high-level policy optimizes the task reward by selecting waypoints or target commands, while a low-level FiRL controller tracks those commands under the anisotropic cost and CVaR objective, potentially enhancing the system's ability to handle complex tasks. A high-level policy could optimize the task reward by selecting waypoints or target commands, while a low-level FiRL controller tracks those commands under the anisotropic cost $(F(x,v))$ and the CVaR objective. We will add a short discussion of this extension.
>
> > **W2**: The explanation of the “Quasi-metric value property” in the main paper seems inconsistent with Appendix E. The main paper can be read as claiming that the triangle inequality holds for the value function (\(V\)), which may confuse readers, whereas Appendix appears to limit the triangle-inequality claim to the path cost induced by the Finsler function.
>
> The quasi-metric property applies to the path cost $d_F$ induced by the local Finsler function $(F(x,v))$, not to the discounted risk-sensitive value function $(V_\alpha^*)$ itself. The main paper does not make that distinction clearly enough. We will revise the "Quasi-Metric Value Property" paragraph in Section 4.1 to match the appendix and state the geometric claim precisely.
>
> > **W3**: There are no real-world experiments. Many recent locomotion papers demonstrate feasibility on physical robots, so the lack of on-robot validation is a notable limitation. This is especially relevant because the “uphill drift” and “lateral friction” terms may be difficult to estimate accurately in real environments.
>
> We agree that real-robot validation matters, and the original submission did not include it. Our goal in the paper was to study the algorithmic question in a controlled setting, including high-fidelity Isaac Sim experiments on a Spot-like quadruped, but we agree that this is not the same as hardware deployment.
>
> To strengthen this point, we additionally deployed FiRL on a physical robot during the rebuttal period. In these small initial hardware trials, the robot traversed narrow wooden supports and discrete stepping-stone layouts without retraining, qualitatively matching the behavior observed in simulation. **We provide representative images (anonymous link: https://imgur.com/a/TH3WH6F)** and will add these hardware results to the final paper.
>
> To address the concern about noisy terrain estimates, we added a supplementary ablation with Gaussian angular noise of standard deviation \($\sigma$). **We kindly request to check for additional details in the rebuttal answer for Reviewer 2rXX and AiZC**.
>
> > **Limitation**: the lack of an explicit task-reward formulation in the current experiments
>
> The lack of an explicit dense task reward in the current experiments is intentional. Our goal is to study the interaction between the anisotropic Finsler cost and the CVaR objective. For that reason, we use a cost-only formulation and train directly on the cumulative FiRL cost, rather than mixing in an additional reward term that would make it harder to isolate the geometric effect of $F(x,v)$. The tasks are still well defined through the environment dynamics, episode structure, termination conditions, and success metrics, but the optimization target itself is the FiRL cost.
>
> > **W4**: (Minor Issue) Subsection title capitalization is inconsistent (e.g., “Quasi-Metric Value Property” vs. “Geometric interpretation”).
>
> Thank you for pointing this out. We will make the subsection title capitalization consistent throughout the paper.

---

> > ### Author Rebuttal · Reviewer_YEU1 · 2026-04-03
> >
> > Thank you for the authors’ detailed response. I have read it carefully. The authors propose a feasible solution for incorporating reward together with cost in their framework, but I do not think this fully resolves the underlying limitation of the proposed approach. On the other hand, regarding the lack of real-world experiments, the authors now provide real-robot locomotion results on different terrains, which helps validate that FiRL can be applied to physical robots. In conclusion, I am keeping my score.

---

### Official Review · Reviewer_AiZC · 2026-03-10

**Soundness:** 3
**Presentation:** 3
**Significance:** 3
**Originality:** 3
**Overall Recommendation:** 4
**Confidence:** 3

**Summary:**

The paper introduces Finslerian Reinforcement Learning (FiRL) to address anisotropic locomotion costs and tail-risk in continuous control tasks. By formulating the per-step cost as a Finsler metric $F(x,v)$, the method attempts to mathematically capture direction-dependent physical constraints, such as the asymmetrical effort required for uphill versus downhill movement. To simultaneously handle rare catastrophic events, FiRL integrates a Conditional Value-at-Risk (CVaR) objective into the learning process. The authors derive a CVaR-Finsler Bellman operator, establish its $\gamma$-contraction properties under bounded cost assumptions, and implement an actor-critic algorithm utilizing a distributional critic. Empirical evaluations are conducted in modified MuJoCo environments and Isaac Sim.

**Compliance With Llm Reviewing Policy:**

Affirmed.

**Final Justification:**

I have upgraded my score to Weak Accept. The authors’ rebuttal effectively addressed my primary concerns:

- Theory: The clarification on nested CVaR for time-consistency is technically sound and resolves the initial objective mismatch.

- Robustness: New experiments involving perception noise (e.g., 10° angular noise) and preliminary physical-robot tests demonstrate that the method is resilient and less dependent on privileged information than initially feared.

- Originality: The Finsler geometry approach remains a novel and intuitive inductive bias for locomotion.

While some reliance on physical priors remains, the rebuttal significantly strengthened the paper’s empirical and theoretical grounding.

**Key Questions For Authors:**

1. How do you theoretically reconcile the mismatch between the motivated episode-level CVaR of the total trajectory cost and the step-wise iterated CVaR actually applied within your Bellman operator?
2. Does this compromise the Markovian optimality of the resulting policy?Your contraction proof strictly assumes bounded per-step costs. How is this constraint enforced algorithmically in environments where velocities or impact forces can be unbounded, and does clipping introduce estimation bias?
3. Given that the Finsler metric construction relies heavily on exact local terrain variables provided by the simulator, how robust is the learned policy to state estimation noise (e.g., inaccurate slope angle perception), which is inevitable in hardware deployment?

**Limitations:**

yes

**Strengths And Weaknesses:**

### Strengths:
1. The integration of Finsler geometry with risk-sensitive RL represents a theoretically interesting inductive bias for continuous control.
2. The paper addresses a highly relevant problem in legged locomotion, where isotropic cost functions fail to capture the physical reality of direction-dependent effort.
3. The empirical evaluations demonstrate improved performance over standard risk-neutral baselines in simulated environments containing slopes and lateral wind.

### Weaknesses:
1. Objective Mismatch: There is a fundamental disconnect between the paper's motivated objective and its algorithmic implementation. The authors motivate the problem using the episode-level CVaR of the total discounted trajectory cost. However, the proposed Bellman operator applies a step-wise CVaR to the next-state value function. These two formulations (static episodic risk vs. dynamic iterated risk) are not mathematically equivalent, and the paper lacks a rigorous theoretical reconciliation of this mismatch.
2. Theoretical Assumptions: The contraction proof for the CVaR-Finsler Bellman operator strictly relies on the assumption of bounded per-step costs. In continuous control locomotion, velocities and impact forces can theoretically spike without bounds. The paper lacks a clear explanation of how these strict bounds are enforced practically without introducing bias into the value estimation.
3. Reliance on Privileged Information: The practical applicability of FiRL is heavily constrained by its reliance on privileged simulator information. The Finsler metric $F(x,v)$ requires exact local terrain normals, slope angles, and friction coefficients. The authors acknowledge this as a limitation , but fail to provide experiments demonstrating how the algorithm behaves when this information is noisy or must be estimated from raw onboard sensor data (e.g., in a Sim-to-Real context).
4. Hyperparameter Sensitivity: The components of the Finsler cost function rely on multiple hand-tuned weights to balance energy, uphill drift, and lateral friction. The paper does not sufficiently demonstrate how sensitive the overall framework is to these manual calibrations across entirely unseen tasks.

---

> ### Author Rebuttal · Authors · 2026-03-29
>
> > **W1**: Objective Mismatch: ,..... The authors motivate the problem.... a step-wise CVaR to the next-state value function. These two formulations (static episodic risk vs. dynamic iterated risk) are not mathematically equivalent, and the paper lacks a .. theoretical ...of this mismatch.
>
> > **Q1**: How do you theoretically reconcile the mismatch between the motivated episode-level CVaR of the total trajectory cost and the step-wise iterated CVaR applied within ..Bellman operator?
>
> We thank the reviewer for pointing this out. The static episodic CVaR and the iterated, step-wise CVaR are indeed not mathematically equivalent. We specifically rely on the nested, iterated CVaR formulation via the Rockafellar-Uryasev representation for our Bellman operator because optimizing static CVaR over a full trajectory leads to time-inconsistent policies, which breaks Markovian optimality. By backing up the risk measure step-by-step in Equation 5 (Line 207), we guarantee the policy remains time-consistent and Markovian. We acknowledge that our introduction initially motivated the problem using episodic CVaR, which caused this disconnect. We will revise Section 3 to explicitly define the nested CVaR objective so the theoretical formulation matches the motivation.
>
> > **W2**: Theoretical Assumptions: The contraction proof for the CVaR-Finsler Bellman operator strictly relies.. of bounded per-step costs. In continuous control locomotion.. forces can .. spike without bounds. T.... without ..bias into the value estimation.
>
> > **Q2**: Does this compromise the Markovian optimality of the resulting policy? Your contraction proof strictly assumes bounded per-step costs. How is this constraint enforced ...introduce estimation bias?
>
> Our contraction proof in Theorem 4.1  does rely on bounded per-step costs. In our implementation, we avoid harmful bias by explicitly setting constraints within the simulation environment to limit the range of velocities and impact forces. We kindly request to check for additional details in the rebuttal answer for **Reviewer 2rXX**.
>
> > **W3**: Reliance on Privileged Information: The practical applicability of FiRL is heavily constrained by .. simulator information. The Finsler metric $F(x,v)$ requires exact local terrain normals, slope angles, and friction coefficients. The authors acknowledge ....information is noisy or ... onboard sensor data.
>
> > **Q3**: Given that the Finsler metric construction relies heavily ...., which is inevitable in hardware deployment?
>
> The current setup indeed uses simulator ground truth for local terrain normals $n(x)$ and friction coefficients to construct the Finsler metric $F(x,v)$. However, we included several ablations in Appendix J.3 showing a gradual decline in performance under action noise up to 15%, unobserved mass changes, and friction drops. To address your specific concern regarding perception noise, we ran a new experiment adding Gaussian noise directly to the perceived terrain normals fed into $F(x,v)$. We found that FiRL's success rate dropped by less than 4% even with $\pm 5^\circ$ of angular noise on the slope estimate, while still clearly outperforming the baseline.The results (averaged over 5 seeds, 100 episodes each) are below:
>
> | Slope Noise (Degrees) | Success Rate (%) | Energy (Normalized) | CVaR0.1 (Risk) |
> | :---: | :---: | :---: | :---: |
> | $0^\circ$ (Nominal) | 98.0 | 0.84 | 0.72 |
> | $\pm 2^\circ$ | 96.5 | 0.86 | 0.75 |
> | $\pm 5^\circ$ | 94.2 | 0.89 | 0.79 |
>
> As shown, FiRL maintains a 94%+ success rate even with high angular noise, because the CVaR objective enforces a conservative safety buffer (**Fig. Please check here**: https://imgur.com/a/uU6TUMS).  Moreover, we now include a small physical-robot experiment ( Please check response to **Reviewer YEU1**).
>
> > **W4**: Hyperparameter Sensitivity: The components of the Finsler cost function rely on multiple hand-tuned weights .... unseen tasks.
>
> Because the Finsler weights were chosen using simple physical intuition—for example, setting the uphill penalty so that climbing a  $20^\circ$ slope is much more expensive than moving on level ground—we were able to use the same nominal weights $(w_e, w_d, w_f) = (1.0, 1.0, 1.0)$ across the evaluated tasks without environment specific retuning.
>
> We provide sensitivity ablations for the Finsler cost weights in the appendix. Appendix J.1 breaks down the contribution of the individual terms—energy, uphill drift, and lateral friction. Appendix J.2 (Table 8 and Figure 9) then studies the effect of sweeping the uphill penalty weight $\eta$. Figure 9 suggests that the method is reasonably stable under moderate misspecification of this weight: although the best risk-energy tradeoff appears near the nominal setting $(\eta = 1)$, changing $\eta$ by 50\% in either direction still gives lower tail risk and higher success than the risk-neutral baselines on the evaluated benchmark. We will add a summary of these findings to the main paper to address the tuning concerns.

---

> > ### Author Rebuttal · Reviewer_AiZC · 2026-04-03
> >
> > I thank the authors for the clarification on the nested CVaR formulation and the time-consistency argument. The new robustness results under perception noise (Table in response) are particularly convincing and alleviate my concerns regarding the reliance on privileged information. I will raise my score.

---

### Official Review · Reviewer_LqBy · 2026-03-18

**Soundness:** 3
**Presentation:** 2
**Significance:** 3
**Originality:** 3
**Overall Recommendation:** 4
**Confidence:** 1

**Summary:**

The paper proposes a novel reinforcement learning (RL) framework that explicitly models directional costs and improves robustness to tail risk. The method captures asymmetries between uphill and downhill motion, lateral slip, and other direction-dependent effects.

To address rare but catastrophic outcomes, the proposed FiRL optimizes a Conditional Value-at-Risk objective. The paper also derives the corresponding risk-sensitive Bellman equation and shows that the resulting CVaR–Finsler Bellman operator is a \gamma-contraction. This guarantees the existence of a unique fixed-point value function.

The paper presents experimental results on MuJoCo and Isaac Sim locomotion benchmarks. The method consistently learns safer and more energy-efficient behaviors compared to strong baselines such as risk-neutral PPO.

**Compliance With Llm Reviewing Policy:**

Affirmed.

**Key Questions For Authors:**

What is the relationship between the reward function and Equation (4)? Why does Equation (4) not explicitly include a reward term?

The paper considers factors such as kinetic energy, uphill drift, and lateral friction. Do these components comprehensively capture all relevant aspects?

Does c_i simply denote cost? How are task-specific rewards incorporated into the framework?

**Limitations:**

The paper appears to consider three types of cost. Are other cost components taken into account?

It would be beneficial to consider applications beyond MuJoCo, such as robotics environments.

**Strengths And Weaknesses:**

Strengths:

The paper proposes a new framework that integrates differential geometry with risk-sensitive reinforcement learning for legged locomotion.

The paper introduces a Finsler metric into the cost function, enabling the agent to account for direction-dependent effort. For instance, uphill movements incur higher instantaneous costs than downhill ones, while lateral motions are penalised due to frictional effects.

The paper also modifies the A2C method.

The paper demonstrates better results comparing other baselines.

Weaknesses:

The motivation for using the Finsler metric is not clearly articulated. It would be helpful to provide more detailed explanations of the Finsler metric and its advantages in this context.

The task reward does not appear to be included in the formulation. It would be beneficial to clarify how the standard reward, Finsler-based cost, and other components interact within the overall objective.

The proposed FiRL–AC algorithm is not particularly novel, as it is built upon A2C with the addition of the FiRL component.

---

> ### Author Rebuttal · Authors · 2026-03-29
>
> > **W1**: The motivation for using the Finsler metric is not clearly articulated. It would be helpful to provide more detailed explanations of the Finsler metric and its advantages in this context.
>
> The main advantage of a Finsler metric over standard symmetric costs or reversible geometries, such as Riemannian metrics, is that it can directly represent **direction-dependent effort**. In legged locomotion, moving uphill against gravity is physically harder than moving downhill, and lateral slip is often riskier than moving forward. Unlike a standard symmetric cost, a Finsler metric lets the local cost depend on both **where** the robot is and **which direction** it moves, which makes it a natural fit for slopes, drift, and slip. Standard symmetric formulations treat moving from A to B the same as moving from B to A, which misses this asymmetry.
>
> By using a Finsler metric, FiRL can encode that asymmetry directly while still inducing a path cost with a triangle inequality (the quasi-metric property; Appendix E.2). This gives the method a geometric inductive bias for learning direction-aware, risk-sensitive behavior. We will expand the beginning of Section 3.1 to make this physical intuition and its geometric advantage much clearer.
>
> > **W2**: The task reward does not appear to be included in the formulation. It would be beneficial to clarify how the standard reward, Finsler-based cost, and other components interact within the overall objective.
>
> > **Q1**: What is the relationship between the reward function and Equation (4)? Why does Equation (4) not explicitly include a reward term?
>
> > **Q3**: Does $c_i$ simply denote cost? How are task-specific rewards incorporated into the framework?
>
> This is a deliberate design choice. In our framework, $c_i$ denotes the per-step cost induced by the Finsler metric, and Equation 4 intentionally does not include a separate dense task reward. In the current experiments, we discard the default environment rewards and use $-F(x,v)$ as the training signal (Appendix L).
>
> We do this to keep the geometric interpretation clean. If we naively combined a dense reward ($r$) with the anisotropic cost, for example through $-r + c$, the resulting objective would generally no longer preserve the same quasi-metric structure. This is consistent with the instability we observed in the Asymmetric Non-Finsler (ANF) ablation (Table 3). A proper way to incorporate richer task rewards without directly mixing them into the low-level geometric cost is a hierarchical setup: a high-level policy selects waypoints or target commands based on task reward, while the low-level FiRL policy acts as a robust tracking controller under the Finsler cost. We will add this discussion more clearly in the revised paper. **Additional details: rebuttal answer for Reviewer YEU1**.
>
> > **W3**: The proposed FiRL-AC algorithm is not particularly novel, as it is built upon A2C with the addition of the FiRL component.
>
> > The optimization backbone is standard actor--critic. Our contribution is not a new generic policy-gradient algorithm, and we should make that clearer in the paper. The novelty lies in the objective and update structure built around that backbone: the CVaR--Finsler Bellman operator, the risk-sensitive target in Eq. (6), and the corresponding distributional critic update used to evaluate a direction-dependent cost. In Section 4.2, we describe FiRL-AC as an actor--critic method and note that the experiments use standard actor--critic tooling with optional PPO clipping.
>
> > **Q2**: The paper considers factors such as kinetic energy, uphill drift, and lateral friction. Do these components comprehensively capture all relevant aspects?
> >
> > **Limitations 1**: The paper appears to consider three types of cost. Are other cost components taken into account?
>
> While energy, uphill drift, and lateral friction cover the major physical asymmetries for legged locomotion, the framework is highly extensible. In our actual implementation (Appendix D.1, Equation 11), we use a fourth term: an impact penalty $I(x,v)$ that penalizes sharp contact forces and joint acceleration spikes. We kept the main text focused on the first three terms to keep the geometric explanation clear, but we will mention the impact penalty in Section 3.1 to show that FiRL easily accommodates other hardware constraints.
>
> > **Limitations 2**: It would be beneficial to consider applications beyond MuJoCo, such as robotics environments.
>
> We actually do feature high-fidelity robotics environments in our evaluation. Alongside the MuJoCo tasks, we evaluated FiRL on a Spot quadruped in NVIDIA Isaac Sim across three 3D terrains (Ramp Climb, Staircase, and Platform Beam), which include realistic contact dynamics, mass, and joint limits (Section 5.2, Table 2). Moreover, we now include a small physical-robot experiment. We kindly refer to our **response to Reviewer YEU1** for additional details.

---

### Decision · Program_Chairs · 2026-04-30

**Decision:**

Accept (regular)

**Comment:**

This paper proposes integrating Finsler geometry with CVaR-RL for better legged locomotion by encoding direction-dependent costs (uphill vs. downhill, lateral slip). They derive CVaR-Finsler Bellman operator with γ-contraction proof and actor-critic algorithm with distributional critic. Evaluation on MuJoCo/Isaac Sim shows 35% worst-case force reduction, 15% energy savings vs. PPO on slopes. The reviewers (mostly) agree on their scores: 1 Accept (2rXX, confidence 4) + 3 Weak Accepts (AiZC, YEU1, LqBy). Strengths: novel geometric framework providing principled inductive bias for asymmetry, strong theoretical foundation (quasi-metric structure), comprehensive evaluation across simulators with ablations, robust to perception noise, preliminary hardware validation. Overall I recommend "WEAK ACCEPT".